



# Exploring how groundwater buffers the influence of heatwaves on vegetation function during multi-year droughts

Mengyuan Mu[1], Martin G. De Kauwe[1], Anna M. Ukkola[1], Andy J. Pitman[1], Weidong Guo[2], Sanaa Hobeichi[1], Peter R. Briggs[3]

[1]ARC Centre of Excellence for Climate Extremes and Climate Change Research Centre, University of New South Wales, Sydney 2052, Australia
[2]School of Atmospheric Sciences and Joint International Research Laboratory of Atmospheric and Earth System Sciences, Nanjing University, Nanjing 210023, China
[3]Climate Science Centre, CSIRO Oceans and Atmosphere, Canberra 2601, ACT, Australia

*Correspondence to*: Mengyuan Mu (mu.mengyuan815@gmail.com)

**Abstract.** The co-occurrence of droughts and heatwaves can have significant impacts on many socioeconomic and environmental systems. Groundwater has the potential to moderate the impact of droughts and heatwaves by moistening the soil and enabling vegetation to maintain higher evaporation, thereby cooling the canopy. We use the Community Atmosphere Biosphere Land Exchange (CABLE) land surface model, coupled to a groundwater scheme, to examine how groundwater influences ecosystems under conditions of co-occurring droughts and heatwaves. We focus specifically on South East Australia for the period 2000–2019 when two significant droughts and multiple extreme heatwave events occurred. We found groundwater plays an important role in helping vegetation maintain transpiration, particularly in the first 1–2 years of a multi-year drought. Groundwater impedes gravity-driven drainage and moistens the root zone via capillary rise. These mechanisms reduced forest canopy temperatures by up to 5°C during individual heatwaves, particularly where the water table depth is shallow. The role of groundwater diminishes as the drought lengthens beyond 2 years and soil water reserves are depleted. Further, the lack of deep roots or stomatal closure caused by high vapour pressure deficit or high temperatures can reduce the additional transpiration induced by groundwater. The capacity of groundwater to moderate both water and heat stress on ecosystems during simultaneous droughts and heatwaves is not represented in most global climate models, suggesting model projections may overestimate the risk of these events in the future.

## 1 Introduction

Droughts and heatwaves are important socio-economic and environmental phenomena, impacting regional food production (Kim et al., 2019; Lesk et al., 2016), water resources (Leblanc et al., 2009; Orth and Destouni, 2018) and the resilience of ecosystems (Ibáñez et al., 2019; Ruehr et al., 2019; Sandi et al., 2020). When droughts and heatwaves co-occur (a "compound event") the consequences can be particularly severe, reducing the terrestrial carbon sink (Ciais et al., 2005), potentially accelerating tree die-off (Allen et al., 2010, 2015; Birami et al., 2018) and setting conditions conducive for wildfires (Jyoteeshkumar reddy et al., 2021). One region experiencing severe coincident heatwaves and drought is Australia (Mitchell et al., 2014). Drought in Australia is associated with large-scale modes of variability, including the El Niño-Southern Oscillation and the Indian Ocean Dipole (van Dijk et al., 2013), and periods of below average rainfall can extend for multiple years (Verdon-Kidd and Kiem, 2009). Heatwaves are commonly synoptically driven, associated with blocking events that can be sustained over many days (Perkins-Kirkpatrick et al., 2016; Perkins, 2015). Modes of variability and synoptic situations are important in setting up conditions conducive to drought and heatwave. However, once a heatwave or drought has become established, land-atmosphere interactions can intensify and prolong both heatwaves and droughts (Miralles et al., 2019), affect their intensity and influence the risk of their co-occurrence (Mukherjee et al., 2020). The role of the land surface in amplifying or dampening heatwaves and droughts is associated with the partitioning of available energy between sensible and latent heat (Fischer et al., 2007; Hirsch et al., 2019) and is regulated by sub-surface water availability (Teuling et al., 2013; Zhou et al., 2019). As soil



moisture becomes more limiting, more of the available energy is converted into sensible heat, reducing evaporative cooling via
latent heat. Changes in the surface turbulent energy fluxes influence the humidity in the boundary layer, the formation of clouds,
incoming solar radiation and the generation of rainfall (D'Odorico and Porporato, 2004; Seneviratne et al., 2010; Zhou et al.,
2019). The sensible heat fluxes warm the boundary layer, leading to heat that can accumulate over several days and exacerbate
heat extremes (Miralles et al., 2014), which can in turn increase the atmospheric demand for water and intensify drought
(Miralles et al., 2019; Schumacher et al., 2019).

Vegetation access to groundwater has the potential to alter these land-atmosphere feedbacks by maintaining vegetation function
during extended dry periods, supporting transpiration and moderating the impact of droughts and heatwaves (Marchionni et al.,
2020; Miller et al., 2010). Where the water table is relatively shallow, capillarity may bring water from the groundwater towards
the surface root zone, increasing plant water availability. Where the water table is deeper, phreatophytic vegetation with tap
roots can directly access groundwater (Zencich et al., 2002). The presence of groundwater, and the access to groundwater by
vegetation is therefore likely to buffer vegetation drought and heatwave stress. For example, groundwater may help vegetation
sustain transpiration and consequently cool plant canopies via evaporation. This is particularly critical during compound events
where cessation of transpiration would increase the risk of impaired physiological function and the likelihood that plants would
exceed thermal limits and risk mortality (Geange et al., 2021; O'sullivan et al., 2017; Sandi et al., 2020).

Quantifying the influence of groundwater on vegetation function has remained challenging as concurrent observations of
groundwater dynamics, soil moisture, and energy and water fluxes are generally lacking over most of Australia and indeed
many parts of the world. Land surface models (LSMs) provide an alternative tool for studying the interactions between
groundwater, vegetation, and surface fluxes in the context of heatwaves and droughts (Gilbert et al., 2017; Martinez et al., 2016a;
Maxwell et al., 2011; Shrestha et al., 2014). However, there has been very little work focused on the influence of groundwater
on droughts and heatwaves occurring at the same time (Keune et al., 2016; Zipper et al., 2019). Our key goal in this paper is
therefore to examine the timescales and extent to which vegetation utilises groundwater during drought and heatwaves, and
determine the degree to which groundwater can mitigate the impacts of compound extremes. We focus on droughts and
heatwaves occurring over south-eastern (S.E.) Australia between during 2000–2019 using the Community Atmosphere
Biosphere Land Exchange (CABLE) LSM. S.E. Australia is an ideal case study since its forest and woodland ecosystems are
known to be dependent on groundwater (Eamus and Froend, 2006; Kuginis et al., 2016; Zencich et al., 2002) and it has
experienced two multi-year droughts and record-breaking heatwaves over the last two decades. By examining the role of
groundwater in influencing droughts and heatwaves, and by understanding how well CABLE can capture the relevant processes,
we aim to build confidence in the simulations of land-atmosphere interactions for future droughts and heatwaves.
**2 Methods**
**2.1 Study area**
The climate over S.E. Australia varies from humid temperate near the coast to semi-arid in the interior. In the last 20 years, S.E.
Australia experienced the 9-year Millennium drought during 2001–2009 (van Dijk et al., 2013) where rainfall dropped from a
climatological average (1970–1999) of 542 mm yr$^{-1}$ to 449 mm yr$^{-1}$, and a 3-year intense recent drought during 2017-2019
where rainfall dropped to 354 mm yr$^{-1}$ (Figure S1). It has also suffered record-breaking summer heatwaves in 2009, 2013, 2017,
and 2019 (Bureau of Meteorology, 2013, 2017, 2019; National Climate Centre, 2009). Here we investigate groundwater
interactions during the period 2000–2019, focusing on the Millennium drought (MD, 2001–2009) and the recent drought (RD,
2017–2019).





**2.2 Overview of CABLE**
CABLE is a process-based LSM that simulates the interactions between climate, plant physiology and hydrology (Wang et al.,
2011). Above ground, CABLE simulates the exchange of carbon, energy and water fluxes, using a single layer, two-leaf
(sunlit/shaded) canopy model (Wang and Leuning, 1998), with a treatment of within-canopy turbulence (Raupach, 1994;
Raupach et al., 1997). CABLE includes a 6-layer soil model (down to 4.6 m) with soil hydraulic and thermal characteristics
dependent on the soil type and soil moisture content. CABLE has been extensively evaluated (e.g., Abramowitz et al., 2008;
Wang et al., 2011; Zhang et al., 2013) and benchmarked (Abramowitz 2012; Best et al. 2015) at global and regional scales.
Here we adopt a version of CABLE (Decker, 2015; Decker et al., 2017) which includes a dynamic groundwater component
with aquifer water storage. This version, CABLE-GW, has been previously evaluated by Decker (2015), Ukkola et al. (2016b)
and Mu et al. (2021) and shown to perform well for simulating water fluxes. CABLE code is freely available upon registration
(https://trac.nci.org.au/trac/cable/wiki); here we use CABLE SVN revision 7765.
**2.3 Hydrology in CABLE-GW**
The hydrology scheme in CABLE-GW solves the vertical redistribution of soil water via a modified Richards equation (Zeng
and Decker, 2009):

$$\frac{\partial \theta}{\partial t} = -\frac{\partial}{\partial z} K \frac{\partial}{\partial z} (\Psi - \Psi_E) - F_{soil} \qquad (1)$$

where $\theta$ is the volumetric water content of the soil (mm$^3$ mm$^{-3}$), $K$ is the hydraulic conductivity (mm s$^{-1}$), $z$ is the soil depth
(mm), $\Psi$ and $\Psi_E$ are the soil matric potential (mm) and the equilibrium soil matric potential (mm), and $F_{soil}$ is the sum of
subsurface runoff and transpiration (mm s$^{-1}$) (Decker, 2015). To simulate groundwater dynamics, an unconfined aquifer is added
to the bottom of the soil column with a simple water balance model:

$$\frac{dW_{aq}}{dt} = q_{re} - q_{aq,sub} \qquad (2)$$

where $W_{aq}$ is the mass of water in the aquifer (mm), $q_{aq,sub}$ is the subsurface runoff in the aquifer (mm s$^{-1}$), and $q_{re}$ is the water
flux between the aquifer and the bottom soil layer (mm s$^{-1}$) computed by the modified Darcy's law:

$$q_{re} = K_{aq} \frac{(\Psi_{aq} - \Psi_n) - (\Psi_{E,aq} - \Psi_{E,n})}{z_{wtd} - z_n} \qquad (3)$$

where $K_{aq}$ is the hydraulic conductivity within the aquifer (mm s$^{-1}$), $\Psi_{aq}$ and $\Psi_{E,aq}$ are the soil matric potentials for the aquifer
(mm), and $\Psi_n$ and $\Psi_{E,n}$ are the soil matric potentials for the bottom soil layer (mm). $z_{wtd}$ and $z_n$ are the depth of the water table
(mm) and the lowest soil layer (mm), respectively. CABLE-GW assumes the groundwater aquifer sits above impermeable
bedrock, giving a bottom boundary condition of:

$$q_{out} = 0 \qquad (4)$$

CABLE-GW computes the subsurface runoff ($q_{sub}$, mm s$^{-1}$) using:

$$q_{sub} = \sin \frac{\overline{d_z}}{d_l} \hat{q}_{sub} e^{-\frac{z_{wtd}}{f_p}} \qquad (5)$$



where $\overline{\frac{d_z}{d_l}}$ is the mean subgrid-scale slope, $\hat{q}_{sub}$ is the maximum rate of subsurface drainage (mm s⁻¹) and $f_p$ is a tunable
parameter. $q_{sub}$ is generated from the aquifer and the saturated deep soil layers (below the third soil layer).
**2.4 Experiment design**
To explore how groundwater influences droughts and heatwaves, we designed two experiments, with and without groundwater
dynamics, driven by the same 3-hour meteorology forcing and land surface properties (see section 2.5 for datasets) for the period
1970-2019. To correct a tendency for high soil evaporation, we implemented a  parameterisation of soil evaporation resistance
that has previously been shown to improve the model (Decker et al., 2017; Mu et al., 2021).
**2.4.1 Groundwater experiment (GW)**
This simulation uses the default CABLE-GW model, which includes the unconfined aquifer to hold the groundwater storage
and simulates the water flux between the bottom soil layer and the aquifer. We first ran the default CABLE-GW with fixed $CO_2$
concentrations at 1969 levels for 90 years by looping the meteorology forcing over 1970–1999. At the end of the 90-year spin-
up, moisture in both the soil column and the groundwater aquifer reached an effective equilibrium when averaged over the study
area. We then ran the model from 1970 to 2019 with time varying $CO_2$. We omit the first 30 years of this period and analyse
the period 2000–2019 to allow for further equilibrium with the time-evolving $CO_2$.
**2.4.2 Free drainage experiment (FD)**
Many LSMs, including those used in the Coupled Model Intercomparison Project 5 (CMIP5), still use a free drainage
assumption and neglect the parameterisation of the unconfined aquifer. To test the impact of this assumption we decoupled the
aquifer from the bottom soil layer and thus removed the influence of groundwater dynamics (experiment FD). In FD, at the
interface between the bottom soil layer and the aquifer, soil water can only move downwards as vertical drainage at the rate
defined by the aquifer hydraulic conductivity:

$q_{re} = K_{aq}$     (6)

This vertical drainage is added to the subsurface runoff flux:

$q_{sub} = q_{sub} + q_{re}$     (7)

The simulated water table depth (WTD) in CABLE-GW affects the water potential gradient between the soil layers via $\Psi_E$
(Zeng and Decker, 2009) and impacts $q_{sub}$ (Equation 5). However, in FD, decoupling the soil column from the aquifer and
adding vertical drainage directly to subsurface runoff causes an artificial and unrealistic decline in WTD. To solve this problem,
we assume a fixed WTD in the FD simulations at 10 m in order to remove this artefact from the simulation of $\Psi_E$ and $q_{sub}$.
The FD simulations are initialized from the near-equilibrated state at the end of the 90-year spin-up used in GW. The period
1970–2019 is then simulated using varying $CO_2$ and the last 20 years are used for analysis.
**2.4.3 Deep root experiment (DR)**
The parameterisation of roots, including the prescription of root parameters in LSMs, is very uncertain (Arora and Boer, 2003;
Drewniak, 2019) and LSMs commonly employ root distributions that are too shallow (Wang and Dickinson, 2012). The vertical
distribution of roots influences the degree to which plants can utilise groundwater, and potentially the role groundwater plays





in influencing droughts and heatwaves. To explore the uncertainty associated with root distribution, we added a "deep root"
(DR) experiment by increasing the effective rooting depth in CABLE for tree areas. In common with many LSMs, CABLE-
GW defines the root distribution following Gale and Grigal (1987):

$f_{root} = 1 - \beta_{root}{}^z$ (8)

where $f_{root}$ is the cumulative root fraction (between 0 and 1) from the soil surface to depth $z$ (m), and $\beta_{root}$ is a fitted parameter
specified for each plant functional type (PFT) (Jackson et al., 1996). In CABLE, the tree areas in our study region are simulated
as evergreen broadleaf PFT with a $\beta_{root} = 0.962$, implying that only 8% of the simulated roots are located below a depth of 64
cm. However, field observations (Canadell et al., 1996; Eberbach and Burrows, 2006; Fan et al., 2017; Griffith et al., 2008)
suggest that the local trees tend to have a far deeper root system, possibly to help cope with the high climate variability. We
therefore increased $\beta_{root}$ for the evergreen broadleaf PFT to 0.99, which assumes 56% of roots are located in depths below
64cm and 21 % of roots below 1.7 m. This enables the roots to extract larger quantities of deep soil water moisture, which is
more strongly influenced by groundwater.

This is a simple sensitivity study, and we therefore only run this experiment during January 2019, when record-breaking
heatwaves compound with the severe recent drought. The DR experiment uses identical meteorology forcing and land surface
properties as GW and FD, and is initialised by the state of the land surface on the 31st December 2018 from the GW experiment.

**2.5 Datasets**

Our simulations are driven by the atmospheric forcing from the Australian Water Availability Project (AWAP), which provides
daily gridded data covering Australia at 0.05° spatial resolution (Jones et al., 2009). This dataset has been widely used to force
LSMs for analysing the water and carbon balances in Australia (Haverd et al., 2013; De Kauwe et al., 2020; Raupach et al.,
2013; Trudinger et al., 2016). The AWAP forcing data include observed fields of precipitation, solar radiation, minimum and
maximum daily temperatures and vapour pressure at 9 am and 3 pm. Since AWAP forcing does not include wind and air pressure
we adopted the near-surface wind speed data from McVicar et al. (2008) and assume a fixed air pressure of 1000 hPa. Due to
missing observations before 1990, the solar radiation input for 1970–1989 was built from the 1990–1999 daily climatology.
Similarly, wind speeds for 1970–1974 are built from the 30-year climatology from 1975 to 2004. We translated the daily data
into 3-hourly resolution using a weather generator (Haverd et al., 2013).

The land surface properties for our simulations are prescribed based on observational datasets. Land cover type is derived from
the National Dynamic Land Cover Data of Australia (DLCD) (https://www.ga.gov.au/scientific-topics/earth-obs/accessing-
satellite-imagery/landcover). We classify DLCD's land cover types to five CABLE PFTs: crop (irrigated/rainfed crop, pasture
and sugar DLCD classes), broadleaf evergreen forest (closed/open/scattered/sparse tree), shrub (closed/open/scattered/sparse
shrubs and open/scattered/sparse chenopod shrubs), grassland (open/sparse herbaceous) and barren land (bare areas). The leaf
area index (LAI) in CABLE is prescribed using a monthly climatology derived from the Copernicus Global Land Service
product (https://land.copernicus.eu/global/products/lai). The climatology was constructed by first creating a monthly time series
by taking the maximum of the 10-daily timesteps each month and then calculating a climatology from the monthly data over
the period 1999–2017. The LAI data was resampled from the original 1 km resolution to the 0.05° resolution following De
Kauwe et al. (2020). Soil parameters are derived from the soil texture information (sand/clay/silt fraction) from SoilGrids (Hengl
et al., 2017) via the pedotransfer functions in Cosby et al. (1984) and resampled from 250 m to 0.05° resolution.





199 To evaluate the model simulations, we use monthly total water storage anomaly (TWSA) at 0.5° spatial resolution from the

200 Gravity Recovery and Climate Experiment (GRACE) and GRACE Follow On products (Landerer et al., 2020; Watkins et al.,

201 2015; Wiese et al., 2016, 2018). The RLM06M release is used for February 2002 – June 2017 and for June 2018 – December

202 2019. We also use the total land evaporation from the Global Land Evaporation Amsterdam Model (GLEAM version 3.5,

203 https://www.gleam.eu/; Martens et al., 2017; Miralles et al., 2011) at 0.5° spatial resolution. For daytime land surface

204 temperature (LST) we use the Moderate Resolution Imaging Spectroradiometer (MODIS) datasets from Terra and Aqua

205 satellites (products MOD11A1 and MYD11A1, Wan and Li, 1997; Wan 2015a, 2015b) at 1 km spatial resolution. We only

206 consider pixels and time steps identified as good quality (QC flags 0). Only the day-time LST values are used due to the lack of

207 good quality night-time LST data. The Terra overpass occurs at 10 am and Aqua at 2 pm local time. To analyse the compound

208 events in January 2019, we linearly interpolate the 3-hourly model outputs to 2 pm to match the overpass time of the Aqua LST.

209 The GRACE, GLEAM and MODIS datasets were resampled to the AWAP resolution using bilinear interpolation.

210

211 To evaluate model performance during heatwaves, we identify heatwave events using the excess heat factor index (EHF, Nairn

212 and Fawcett, 2014). EHF is calculated using the daily AWAP maximum temperature, as the product of the difference of the

213 previous 3 day mean to the 90th percentile of the 1970–1999 climatology and the difference of the previous 3 day mean to the

214 preceding 30 day mean. A heatwave occurs when the EHF index is greater than 0 for at least three consecutive days. We only

215 focus on summer heatwaves occurring between December and February of the following year.

216

### 217 3 Results

### 218 3.1 Simulations for the Millennium Drought and the recent drought

219 Previous studies have shown that simulations by LSMs diverge as the soil dries (Ukkola et al., 2016a), associated with

220 systematic biases in evaporative fluxes and soil moisture states in the models (Mu et al., 2021; Swenson and Lawrence, 2014;

221 Trugman et al., 2018). We therefore first evaluate how well CABLE-GW captures the evolution of terrestrial water variability

222 during two recent major droughts.

223

224 Figure 1a shows the total water storage anomaly during 2000–2019 observed by GRACE and simulated in GW and FD. Both

225 GW and FD accurately capture the interannual variability in total water storage for S.E. Australia (r = 0.96 in GW, and 0.90 in

226 FD). Both model configurations simulate a decline in TWSA through the first drought period (up to 2009, see Figure S1), the

227 rapid increase in TWSA from 2010 associated with higher rainfall, a decline from around 2012 due to the re-emergence of

228 drought conditions, and the rapid decline during the recent drought after conditions had eased in 2016 (Figure S1). FD

229 underestimates the magnitude of monthly TWSA variance (standard deviation, SD = 37.18 mm) compared to GRACE (47.74

230 mm) or GW (47.67 mm). This underestimation is linked with the lack of aquifer water storage in the FD simulations which

231 provides a reservoir of water that changes slowly and has a memory of previous wet/dry climate conditions (Figure 1a).

232

233 Figure 1b shows the accumulated precipitation (P) minus evaporation (E) over the two drought periods. GW increases the

234 evaporation relative to FD such that the accumulated P−E decreases from about 786 mm to 455 mm during the Millennium

235 drought, which is much closer to the GLEAM estimate (97 mm). A similar result, although over a much shorter period, is also

236 apparent for the recent drought (Figure 1b). The lower P−E in GW suggests that the presence of groundwater storage alleviates

237 the vegetation water stress during droughts, and reduces the reliance of E on P, indicated by a small reduction in the correlation

238 (r) between E and P from 0.28 in FD to 0.24 in GW for MD, and a reduction from 0.42 to 0.37 for RD (Figure 1b). The GW

239 simulations are also closer to the GLEAM estimates which suggests that adding groundwater improves the simulations during



droughts. The difference in E is also demonstrated spatially in Figure S2. During the Millennium drought, the GW simulations
show a clear improvement over FD in two aspects. GW shows smaller biases in E along the coast where FD underestimates E
strongly (Figure S2b-c). The areas where E is underestimated are also smaller in extent in GW, suggesting that GW overall
reduces the dry bias. The magnitude of the bias in GW reaches around 300 mm over small areas of S.E. Australia while in the
FD simulations biases are larger, reaching 400 mm over a larger area. Overall, Figure 1 and Figure S2 show that representing
groundwater improves the simulation of the inter-annual variability in the terrestrial water cycle and storage, particularly during
droughts.

**3.2 The role of groundwater in sustaining evaporation during droughts**

We next explore the mechanisms by which including groundwater modifies the simulation of evaporation. Figure 2 displays the
overall influence of groundwater on water fluxes during the recent drought. GW simulates 50–200 mm yr$^{-1}$ more E over coastal
regions where there is high tree cover (Figure 2a; see Figure S3 for land cover). Adding groundwater also increases E in most
other regions, although the impact is negligible in many inland and non-forested regions (i.e., west of 145ºE). We identified a
clear connection between E (Figure 2a) and the simulated WTD in the GW simulations (Figure S4). GW simulates 110 mm yr$^{-1}$
more E when the WTD is shallower than 5 m deep, 22 mm yr$^{-1}$ when the WTD is 5–10 m deep, but only 3 mm yr$^{-1}$ more when
the WTD is below 10 m. Higher transpiration ($\Delta Et$) in GW explains 78% of the total evaporative difference between GW and
FD where WTD is shallower than 5 m (Figure 2b). This is confirmed by the change in the soil evaporation ($\Delta Es$) (Figure 2c)
where adding groundwater increases Es by negligible amounts over most of S.E. Australia, but by up to 25 mm y$^{-1}$ in regions
underlain by shallow groundwater (Figure S4), which is consistent with field observations that indicate that Es can be substantial
under conditions of a very shallow water table (Thorburn et al., 1992). In the very shallow WTD areas, the excess Es in GW
results from the capillary rise of moisture from the shallow groundwater to the surface.
A significant factor in explaining how groundwater influences E is through changes in drainage and recharge from the aquifer.
Figure 2d shows that the vertical drainage (Dr) both increases and decreases depending on the location. The addition of
groundwater reduces vertical drainage by 74 mm yr$^{-1}$ where WTD is shallower than 5 m. In some regions, the drainage increases
with the inclusion of groundwater by up to 100 mm yr$^{-1}$, especially in the areas where WTD is ~ 5 m. This is associated with
the WTD being slightly below the bottom of the soil column (4.6 m). When the groundwater aquifer is nearly full and the bottom
soil layer is relatively wet, the calculated hydraulic conductivity ($K_{aq}$) in GW is much larger than in FD where the bottom soil
layer is drier due to a lack of groundwater contribution. This leads to higher vertical drainage in GW and a positive $\Delta Dr$. Inland,
where the WTD tends to be much deeper there is no significant difference in Dr between GW and FD.
Figure 2e shows the difference in recharge into the upper soil column ($\Delta Qrec$) between GW and FD. The recharge from the
aquifer into the bottom soil layer provides 17 mm yr$^{-1}$ extra moisture in the GW simulations in regions with a WTD between
5–10 m and 10 mm yr$^{-1}$ where the WTD is deeper than 10 m, helping to explain the changes in E and Et in areas with deep
WTD. However, there is no significant $\Delta Qrec$ in regions with a shallow WTD (~5 mm yr$^{-1}$), suggesting the influence of
groundwater is mainly via reduced drainage in these locations. Recharge can only occur when WTD is below the soil column
(bottom boundary at 4.6m depth). If WTD is shallow and within the soil column, the interface is saturated and no recharge from
the aquifer to the soil column can occur and water only moves downwards by gravity.
The combined impact of reduced drainage in GW (Figure 2d) and recharge into the root-zone (Figure 2e) is an increased water
potential gradient between the drier top soil layers and the wetter deep soil layers, encouraging overnight capillary rise. Taking
the hot and dry January 2019 as an example, when the compound events occurred, Figure 2f shows the maximum water stress
factor difference ($\Delta\beta$) overnight (between 9 pm and 3 am, i.e. predawn when soil is relatively moist following capillary lift



overnight). We only consider rainless nights to exclude the impact of drainage induced by precipitation. The water stress factor
($\beta$) is based on the root distribution and moisture availability in each soil layer and represents the soil water stress on transpiration
as water becomes limiting. Figure 2f implies that while the redistribution of moisture is small overall, in some locations it can
reduce moisture stress by up to 4–6%.
**3.3 The impact of groundwater during heatwaves**
We next explore whether the higher available moisture due to the inclusion of groundwater enables the canopy to cool itself via
evaporation during heatwaves by examining the temperature difference between the simulated canopy temperature ($T_{canopy}$, ºC)
and the forced air temperature ($T_{air}$, ºC). We focus on the forested regions (Figure S3) as the role of groundwater in enhancing
plant water availability was shown to be largest in these regions (Figure 2).

Figure 3a shows the average $T_{canopy}$−$T_{air}$ ($\Delta T$, ºC) over the forested regions for summer heatwaves from the GW and FD
simulations, with the grey line indicating the median $\Delta T$ difference. During heatwaves, the inclusion of groundwater moistens
the soil and supports higher transpiration, cooling the canopy and reducing $\Delta T$ relative to FD by up to 0.76ºC (e.g. January
2013). As the drought lengthens in time (Figure 1a), the depletion of moisture gradually reduces this effect. The impact of
groundwater is clear in the evaporative fraction (Figure 3b) where in periods of higher rainfall (e.g. 2010–2011; Figure S1), and
at the beginning of a drought (2001, 2017), the EF is higher (0.03 to 0.18). This implies more of the available energy is
exchanged with the atmosphere in the form of latent, rather than sensible heat. However, the strength of the cooling effect
decreases as the droughts extends, because the vegetation becomes increasingly water-stressed which consequently limits
transpiration (Figure 3c).

Figure 4a shows the spatial map of $\Delta T$ simulated in GW during heatwaves in the 2017–2019 drought. It indicates both land
cover type (Figure S3) and WTD (Figure S4) contribute to the $\Delta T$ pattern. The evaporative cooling via transpiration is stronger
over the forested areas compared to crop or grassland, and stronger in the regions with a wetter soil associated with a shallower
WTD. However, EF is mainly determined by WTD (compare Figure 4b and Figure S4). Inland, where the WTD is deeper and
the soil is drier, most of the net radiation absorbed by the land surface is partitioned into sensible rather than latent heat (Figure
4b). However, in the coastal regions with a shallow WTD, the wetter soil reduces the water stress (Figure 4c), enables a higher
EF (Figure 4b), and alleviates heat stress on the leaves (Figure 4a). Along the coast where WTD is shallow, GW simulates a
cooler canopy temperature due to the higher evaporative cooling (Figure 4e) which is the consequence of a lower soil water
stress (Figure 4f) linked to the influence of groundwater (Figure S4).

Figure 5 shows the density scatter plot of $\Delta T$ versus WTD in S.E. Australia forested areas during heatwaves in 2000–2019. A
shallow WTD moderates the temperature difference between the canopy and the ambient air during heatwaves leading to a
smaller temperature difference. Meanwhile, as the WTD increases, due to the limited rooting depth in the model, the ability of
the groundwater to support transpiration and offset the impact of high air temperatures is reduced. Figure 5 shows a large amount
of variations, but nonetheless implies a threshold of ~6 m whereafter there is a decoupling and little influence from groundwater
during heatwaves.
**3.4 The impact of groundwater during the drought and heatwave compound events**
To examine the influence of groundwater on heatwaves occuring simultaneously with drought, we focus on a case study of the
record-breaking heatwaves in January 2019, which is the hottest month on record for the study region (Bureau of Meteorology,
2019). The unprecedented prolonged heatwave period started in early December 2018 and continued through January 2019 with
three peaks. We select two days (15th and 25th January 2019), when heatwaves spread across the study region, from the second





and third heatwave phases (Figure S5).

We evaluate CABLE $T_{canopy}$ against MODIS LST observations, concentrating on forested areas where MODIS LST should
more closely reflect vegetation canopy temperatures, but note that this comparison is not direct as the satellite estimate will
contain contributions from the understorey and soil. Figures 6a-b show the good quality MODIS LST minus $T_{air}$ at 2 pm
($\Delta T_{MOD\_2pm}$) over forested regions on the 15th and 25th January 2019, and Figures 6c-d display the matching GW-simulated $\Delta T$
at 2 pm ($\Delta T_{GW\_2pm}$). Overall, $\Delta T_{GW\_2pm}$ increases from the coast to the interior in both heatwaves, consistent with the $\Delta T_{MOD\_2pm}$
pattern in both heatwaves, albeit that $\Delta T_{GW\_2pm}$ appears to be biased high relative to $\Delta T_{MOD\_2pm}$ along the coastal forests (Figure
S6a-b).

Figure 6e-f shows the $\Delta T_{2pm}$ difference between GW and FD. Access to groundwater can reduce canopy temperature by up to
5°C, in particular when the WTD is shallow. While reductions of 5°C are clearly limited in spatial extent, the overall pattern of
cooling associated with groundwater access is quite widespread implying a reduction in heat stress experienced by the woody
vegetation during heatwaves. Generally, GW matches MODIS LST better than FD despite the bias in both simulations (compare
Figure S6 a-b and Figure S6 c-d). Nevertheless, the temperature reduction between GW and FD is still modest (< 1°C) for most
of the forested regions. This may be related to the shallow root distribution assumed in many LSMs, which prevents roots from
directly accessing the moisture stored in the deeper soil (note, CABLE assumes 92% of all roots are in the top 64 cm). To
examine this possibility, we performed the deep root (DR) sensitivity experiment which prescribed more roots in the deeper soil
(56% below 64cm depth). Figure S6e-f illustrates the difference between $\Delta T_{2pm}$ in DR and $\Delta T_{GW\_2pm}$. By enabling access to
moisture in the deeper soil, the LSM simulates further cooling by 0.2–5°C across the forests. The prescribed deeper roots also
lead to an overall better simulation of $\Delta T$ at 2 pm relative to the MODIS LST (compare Figure S6g-h with Figure S6a-b).

Figure 6g-h shows the diurnal cycles of $\Delta T$ for the two selected regions (red boxes in Figure 6e-f) compared with the MODIS
LST estimated. The region highlighted for the 15th January (Figure 6g) has a WTD of 4–7 m, while the region highlighted for
the 25th January (Figure 6h) has a WTD < 4m (Figure S4). In both regions, the simulated $\Delta T$ is highest in FD, lower in GW and
lowest in DR. Where the WTD is 4–7m (Figure 6g), the three simulated $\Delta T$ are slightly lower than $\Delta T$ calculated by MODIS
LST (red squares). However, in the shallower WTD region (Figure 6h), the simulated $\Delta T$ between experiments is more dispersed
across experiments and exceeds the MODIS $\Delta T$ at both time points, implying that neglecting groundwater dynamics and deep
roots is more likely to cause an overestimation of heat stress in the shallower WTD region.
**3.5 Constraints on groundwater mediation during the compound events**
We finally probe the reasons for the apparent contradiction between the large impact of groundwater on E during drought
(Figure 2a) but a smaller impact on $\Delta T$ during the compound events (Figure 6e-f). Figure 7 shows three factors ($\beta$, vapour
pressure deficit (D) and $T_{air}$) that constrain the impact of groundwater on $\Delta T$ in CABLE during heatwaves in January 2019.
Figure 7a shows the difference in $\Delta T$ between GW and FD as a function of $\Delta\beta$, suggesting that the inclusion of groundwater
has a large impact on $\Delta T$ when there is a coincidental and large difference in $\beta$ between the GW and FD simulations. Figure 7b
indicates a clear threshold at D = 3 kPa where GW and FD converge, while Figure 7c shows a convergence threshold when the
$T_{air}$ exceeds 32°C. Above these two thresholds, access to groundwater seemingly becomes less important in mitigating plant
heat stress. There are two mechanisms in CABLE that explain this behaviour. First, as D increases, CABLE predicts that stomata
begin to close following a square root dependence (De Kauwe et al., 2015; Medlyn et al., 2011). Second, as $T_{air}$ increases,
photosynthesis becomes inhibited as the temperature exceeds the optimum for photosynthesis. In both instances, evaporative
cooling is reduced, regardless of the root zone moisture state dictated by groundwater access. That is to say, access to
groundwater has limited capacity to directly mediate the heat stress on plants during a compound event when the air is very dry,



or very hot.

## 4 Discussion

In the absence of direct measurements, we used the CABLE-GW LSM, constrained by satellite observations to investigate how
groundwater influences ecosystems under conditions of co-occurring droughts and heatwaves. We found that representing
groundwater was most important during the onset of drought and the first ~two years of a multi-year drought. This primarily
occurred via impeding gravity-driven drainage (Figure 2d) but also via capillary rise from the groundwater aquifer (Figure 2e).
This moistening enabled the vegetation to sustain higher E for at least a year (Figure S7).

When a heatwave occurs during a drought, and in particular early in a drought, the extra transpiration enabled by representing
groundwater dynamics helps reduce the heat stress on vegetation (e.g. the reduction of 0.64°C of $\Delta T$ over the forests in 2002,
Figure 3a). This effect is particularly pronounced in regions with a shallower WTD (e.g. where the groundwater was within the
first 5m, there was a 1°C reduction in $\Delta T$ in the recent drought, Figure 4d). Importantly, the role played by groundwater
diminishes as the drought lengthens beyond two years. Additionally, either the lack of deep roots or stomatal closure caused by
high D/ $T_{air}$ can reduce the additional transpiration induced by groundwater. The latter plant physiology feedback dominates
during heatwaves co-occurring with drought, even if the groundwater's influence has increased root-zone water availability.

Our results highlight the impact of groundwater on both land surface states (e.g. soil moisture) and on surface fluxes and how
this impact varies with the length and intensity of droughts and heatwaves. The results imply that the dominant mechanism by
which groundwater buffered transpiration was through impeding gravity-driven drainage. We found a limited role for upward
water movement from aquifer due to simulated shallow WTD (which was broadly consistent with the observations in Fan et al.,
2013). Further work will be necessary to understand how groundwater interacts with droughts and heatwaves and what these
interactions mean for terrestrial ecosystems and the occurrence of the compound extreme events, particularly under the
projection of intensifying droughts (Ukkola et al., 2020) and heatwaves (Cowan et al., 2014).

### 4.1 Changes in the role of groundwater in multi-year droughts

Groundwater is the slowest part of the terrestrial water cycle to change (Condon et al., 2020) and can have a memory of multi-
year variations in rainfall (Martínez-de la Torre and Miguez-Macho, 2019; Martinez et al., 2016a). Our results show that the
effect of groundwater on the partitioning of available energy between latent and sensible heat fluxes is influenced by the length
of drought. As the drought extends in time, the extra E sustained by groundwater decreases (e.g. during the Millennium drought,
Figure S7). The role of a drying landscape in modifying the partitioning of available energy between latent and sensible heat
fluxes is well known and has been extensively studied (Fan, 2015; Miralles et al., 2019; Seneviratne et al., 2010). Our results
add to the knowledge by quantifying the extent of the groundwater control, and eliciting the timescales of influence and the
mechanisms at play. The importance of vegetation-groundwater interactions on multi-year timescales has been identified
previously. Humphrey et al. (2018) hypothesised that climate models may underestimate the amplitude of global net ecosystem
exchange because of a lack of deep-water access. Our regional based results support this hypothesis and in particular highlight
the importance of groundwater for explaining the amplitude of fluxes in wet regions (Figure 1), as well as sustaining evaporation
during drought.

### 4.2 Implications for land-atmosphere feedbacks during compound events

Our results show that during drought-heatwave compound events, the existence of groundwater eases the heat stress on the
forest canopy and reduces the sensible heat flux to atmosphere. This has the potential to reduce heat accumulating in the



boundary layer and help ameliorate the intensity of a heatwave (Keune et al., 2016; Zipper et al., 2019). The presence of
groundwater helps dampen a positive feedback loop whereby during drought-heatwave compound events, the high exchange of
sensible and low exchange of latent heat can heat the atmosphere and increase the atmospheric demand for water (De Boeck et
al., 2010; Massmann et al., 2019), intensifying drying (Miralles et al., 2014). The lack of groundwater in many LSMs suggests
a lack of this moderating process and consequently a risk of overestimating the positive feedback in coupled climate simulations.
Our results show that neglecting groundwater leads to an average overestimate in canopy temperature by 0.2–1°C where the
WTD is shallow (Figure 4d), but as much as 5°C in single heatwave events (Figure 6e-f), leading to an increase in the sensible
heat flux (Figure 4e).

The capacity of groundwater to moderate this positive land-atmosphere feedback is via modifying soil water availability. Firstly,
soil water availability influenced by WTD affects how much water is available for E. In the shallow WTD regions, the higher
soil water is likely to suppress the mutual enhancement of droughts and heatwaves (Keune et al., 2016; Zipper et al., 2019),
particularly early in a drought. However, this suppression becomes weaker as the WTD deepens, in particular at depths beneath
the root zone (e.g. 4.6 m in CABLE-GW) or as a drought lengthens. Our results imply the land-amplification of heatwaves is
likely stronger in the inland regions (Hirsch et al., 2019) where the WTD is lower than 5m and the influence of groundwater
diminishes (Figure S4), and once a drought has intensified significantly.

On a dry and hot heatwave afternoon, plant physiology feedbacks to high D and high $T_{air}$ dominate transpiration and reduce the
influence of groundwater in moderating heatwaves. In CABLE, stomatal closure occurs either directly due to high D (>3 kPa)
(De Kauwe et al. 2015) or indirectly due to biochemical feedbacks on photosynthesis at high $T_{air}$ (>32°C) (Kowalczyk et al.,
2006); both processes reduce transpiration to near zero, eliminating the buffering effect of groundwater on canopy temperatures.
While the timing of the onset of these physiology feedbacks varies across LSMs due to different parameterised sensitivities of
stomatal conductance to atmospheric demand (Ball et al., 1987; Leuning et al., 1995) and different temperature dependence
parameterisations (Badger and Collatz, 1977; Bernacchi et al., 2001; Crous et al., 2013), importantly, stomatal closure during
heat extremes would be model invariant.

### 4.3  Uncertainties and future directions

Our study uses a single LSM and consequently the parameterisations included in CABLE-GW influence the quantification of
the role of groundwater on droughts and heatwaves. We note CABLE-GW has been extensively evaluated for water cycle
processes (Decker, 2015; Decker et al., 2017; Mu et al., 2021; Ukkola et al., 2016b), but evaluation for groundwater interactions
remains limited due to the lack of suitable observations (e.g. regional WTD monitoring or detailed knowledge of the distribution
of root depths). Figure 1 gives us confidence that CABLE-GW is performing well, based on GRACE data, and other evaluation
of CABLE highlights the capacity of CABLE-GW to simulate E well (Decker et al., 2017) but we note here key model
parameterisations that may influence the role of groundwater are particularly uncertain.

We need to be cautious about the "small" groundwater impact on the canopy temperature and associated turbulent energy fluxes
during high D or high $T_{air}$ (Figure 3, 4, 6). The thresholds of D and $T_{air}$ currently assumed by LSMs are in fact likely to be
species specific. Australian trees in particular have envolved a series of physiological adaptations to reduce the negative impact
of heat extremes. It is important to note that most LSMs parameterise their stomatal response to VPD for moderate ranges (< 2
kPa), which leads to significant biases at high D (Yang et al., 2019), a feature common in Australia and during heatwaves in
general. New theory is needed to ensure that models adequately capture the full range of stomatal response to variability in D
(low and high ranges). Similarly, while there is strong evidence to suggest that the optimum temperature for photosynthesis
does not vary predictably with the climate of species origin (Kumarathunge et al., 2019) (implying model parameterisations do



not need to vary with species), findings from studies do vary (Cunningham and Reed 2002; Reich et al. 2015). Moreover,
evidence that plants acclimate their photosynthetic temperature response is strong (Kattge and Knorr, 2007; Kumarathunge et
al., 2019; Mercado et al., 2018; Smith et al., 2016; Smith and Dukes, 2013). As a result, it is likely that LSMs currently
underestimate groundwater influence during heatwaves due to the interaction with plant physiology feedbacks. This is a key
area requiring further investigation. For example, Drake et al. (2018) demonstrated that during a 4-day heatwave > 43°C,
Australian *Eucalyptus parramattensis* trees did not reduce transpiration to zero as models would commonly predict, allowing
the trees to persist unharmed in a whole-tree chamber experiment. Although De Kauwe et al. (2019) did not find strong support
for this phenomenon across eddy covariance sites, if this physiological response is common across Australian woodlands, it
would change our view on the importance of soil water availability (therefore groundwater) on the evolution of heatwave or
even compound events. Coupled model sensitivity experiments may be important to determine the magnitude that such a
physiological feedback would present and could guide the direction of future field/manipulation experiments.
Root distribution and root function and thereby how roots utilise groundwater are uncertain in models (Arora and Boer, 2003;
Drewniak, 2019; Wang et al., 2018; Warren et al., 2015) and indeed in observations (Fan et al., 2017; Jackson et al., 1996;
Schenk and Jackson, 2002). Models often ignore how roots forage for water and respond to moisture heterogeneity, limiting the
model's ability to accurately reflect the plant usage of groundwater (Warren et al., 2015). In LSMs, roots are typically
parameterised using a fixed distribution and normally ignore water uptake from deep roots. This assumption neglects any
climatological impact of root distribution and the differentiation in root morphology and function (fine roots vs tap roots),
leading to a potential underestimation of groundwater utilization in LSMs (see our deep root experiment, Figure 6g-h). This
assumption may be particularly problematic in Australia where vegetation has developed significant adaptation strategies to
cope with both extreme heat and drought, including deeply rooted vegetation that can access groundwater (Bartle et al., 1980;
Dawson and Pate, 1996; Eamus et al., 2015; Eberbach and Burrows, 2006; Fan et al., 2017). We also note that CABLE does
not directly consider hydraulic redistribution, defined as the passive water movement via plant roots from moister to drier soil
layers (Burgess et al., 1998; Richards and Caldwell, 1987). Neglecting hydraulic redistribution has the potential to underestimate
the groundwater transported upwards and understate the importance of groundwater on ecosystems.
On the atmosphere side, the existence of groundwater increases the water flux from the land to atmosphere, particularly in
regions of shallow WTD, during the first 1–2 years of a drought. This has the potential to moisten the lower atmosphere and
may encourage precipitation (Anyah et al., 2008; Jiang et al., 2009; Martinez et al., 2016b; Maxwell et al., 2011). However, our
experiments are uncoupled from the atmosphere so while there is the potential for the higher E to affect the boundary layer
moisture (Bonetti et al., 2015; Gilbert et al., 2017; Maxwell et al., 2007), clouds and precipitation, we cannot conclude that it
would until we undertake future coupled simulations.
Finally, we note we have focused on the role of groundwater in a natural environment. Humans extract large quantities of
groundwater in many regions (Döll et al., 2014; Wada, 2016). Adding human management of groundwater into LSMs enables
an examination of how this affects the vulnerability of ecosystems to heatwaves and drought, and may ultimately identify those
vulnerable ecosystems close to tipping points that are priorities for protection.
**5 Conclusions**
In conclusion, we used the CABLE LSM, constrained by satellite observations, to explore the timescales and extent to which
groundwater influences vegetation function and turbulent energy fluxes during multi-year droughts. We showed that
groundwater moistened the soil during the first ~two years of a multi-year drought which enabled the vegetation to sustain



higher evaporation (50–200 mm yr$^{-1}$ over the coastal forest regions). This cooled the forest canopy on average by 0.03–0.76 ℃
and as much as 5 ℃ in regions of shallow water table depths, helping to moderate the heat stress on vegetation during heatwaves.
However, the ability of groundwater to buffer vegetation function varied with the length and intensity of droughts and heatwaves,
with its influence decreasing with prolonged drought conditions. Importantly, we also demonstrated that the capacity of the
groundwater to buffer evaporative fluxes during heatwaves is dependent on plant physiology feedbacks which regulate stomatal
control, irrespective of soil water status. Given increased risk of regional heatwaves and droughts in the future, the role of
groundwater on land-atmosphere feedbacks and on terrestrial ecosystems needs to be better understood in order to constrain
future projections.
**Code and data availability**
The CABLE code is available at https: //trac.nci.org.au/trac/cable/wiki (last access: 30 April 2021) (NCI, 2021) after registration.
Here, we use CABLE revision r7765. Scripts for plotting and processing model outputs are available at
https://github.com/bibivking/Heatwave/tree/master/GW_DH. GRACE land is available at http://grace.jpl.nasa.gov, supported
by the NASA MEaSUREs Program. GLEAM dataset is available at https://www.gleam.eu/. MOD11A1 MODIS/Terra Land
Surface Temperature and the Emissivity Daily L3 Global 1km and MYD11A1 MODIS/Aqua Land Surface Temperature and
the Emissivity Daily L3 Global 1km datasets were acquired from the NASA Land Processed Distributed Active Archive Center
(LP DAAC), located in the USGS Earth Resources Observation and Science (EROS) Center in Sioux Falls, South Dakota, USA
(https://lpdaacsvc.cr.usgs.gov/appeears/).

**Author contributions**
MM, MGDK, AJP and AMU conceived the study, designed the model experiments, investigated the simulations and drafted
the manuscript. SH and PRB provided the evaluation and the meteorology forcing datasets. All authors participated in the
discussion and revision of the manuscript.

**Acknowledgements**
Mengyuan Mu, Martin G. De Kauwe, Andy J. Pitman, Anna M. Ukkola and Sanaa Hobeichi acknowledge support from the
Australian Research Council (ARC) Centre of Excellence for Climate Extremes (CE170100023). Mengyuan Mu acknowledges
support from the UNSW University International Postgraduate Award (UIPA) scheme. Martin G. De Kauwe and Andy J. Pitman
acknowledge support from the ARC Discovery Grant (DP190101823) and Anna M. Ukkola support from the ARC Discovery
Early Career Researcher Award (DE200100086). Martin G. De Kauwe acknowledges support from the NSW Research
Attraction and Acceleration Program (RAAP). We thank the National Computational Infrastructure at the Australian National
University, an initiative of the Australian Government, for access to supercomputer resources. Mengyuan Mu thanks the
University of Nanjing for hosting her research through 2020.

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






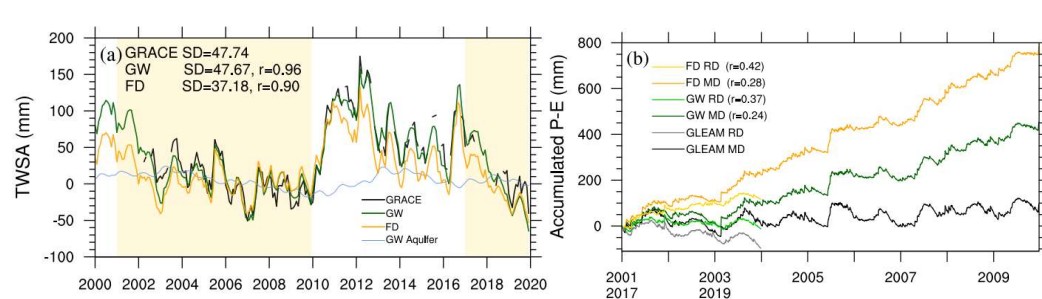



**Figure 1** (a) Total water storage anomaly (TWSA) during 2000–2019 and (b) accumulated P–E for the two droughts over S.E. Australia. In
panel (a), observations from GRACE are shown in black, the GW simulation in green, FD in orange and the aquifer water storage anomaly in
GW in blue. The shading in panel (a) highlights the two drought periods. The left top corner of panel (a) displays the correlation (r) between
GRACE and GW/FD, as well as the standard deviation (SD, mm) of GRACE, GW and FD over the periods when GRACE and the simulations
coincide. Panel (b) shows the accumulated P–E for two periods; the dark lines show the 2001–2009 Millennium drought (MD) and the light
lines show the 2017-2019 recent drought (RD). The correlation (r) between the P and E is shown in the legend of panel (b).










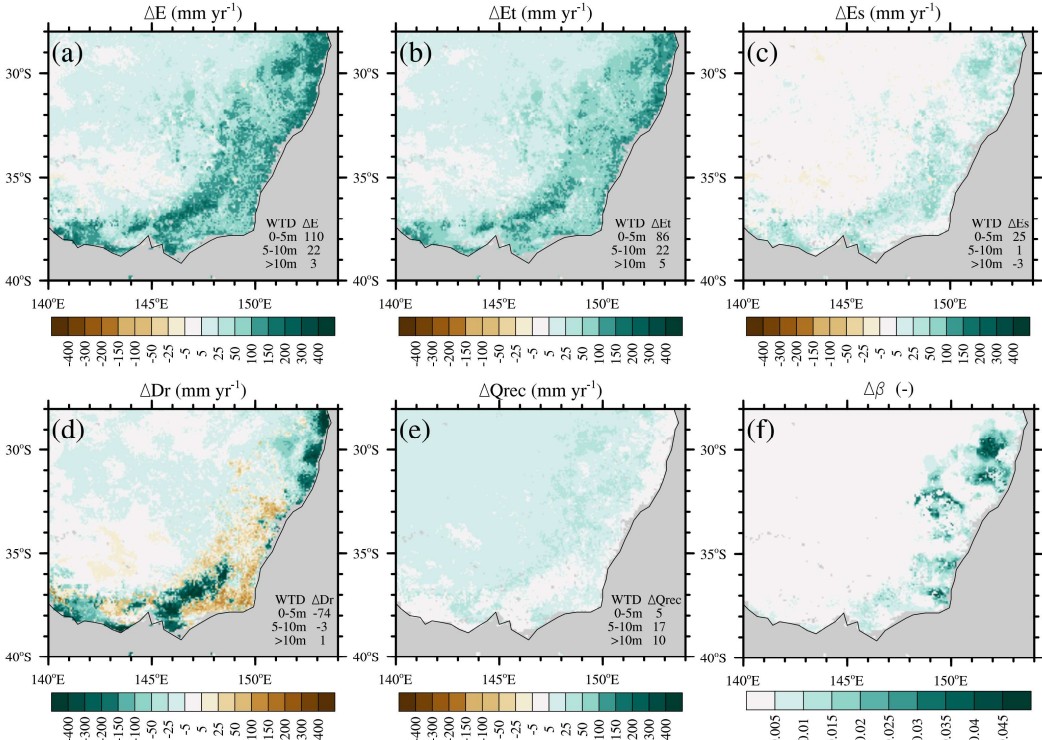


**Figure 2.** The overall influence of groundwater during the recent drought. (a)-(e) are the difference (GW−FD) in total evaporation (ΔE),
transpiration (ΔEt), soil evaporation (ΔEs), vertical drainage (ΔDr) and recharge (ΔQrec), respectively. In the bottom right of panels (a)-(e),
the average of each variable over selected water table depths (WTD) is provided. (f) is the maximum night-time water stress factor difference
(Δβ) between 3 am (i.e. predawn when the soil is relatively moist following capillary lift overnight) and 9 pm the previous day. We only
include rainless nights in January 2019 to calculate Δβ to remove any influence of overnight rainfall.

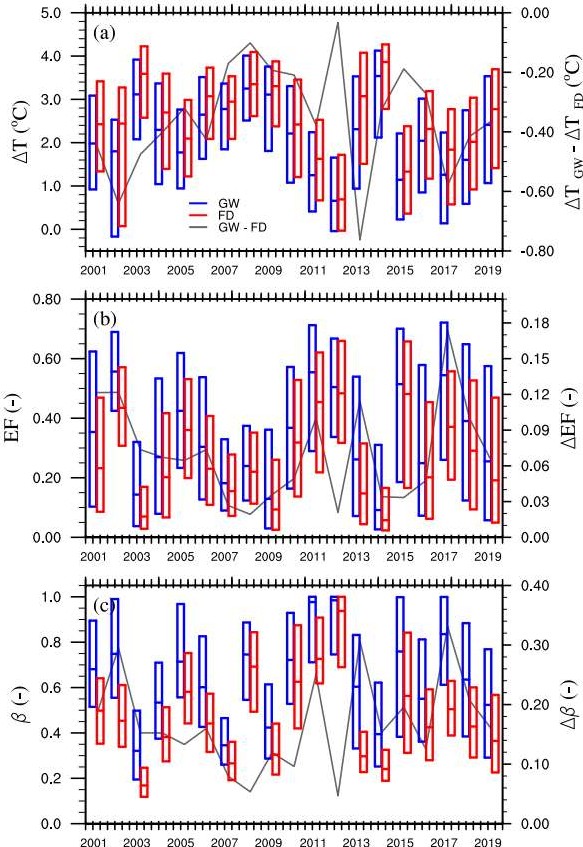

871

**Figure 3.** Groundwater-induced differences in (a) $T_{canopy}-T_{air}$ ($\Delta T$), (b) evaporative fraction (EF) and (c) water stress factor ($\beta$) during 2000–
2019 summer heatwaves over forested areas. The left y-axis is the scale for boxes. The blue boxes refer to the GW experiment and the red
boxes to FD. For each box, the middle line is the median, the upper border is the 75th percentile, and the lower border is 25th percentile. The
right y-axis is the scale for the grey lines which display the difference in the medians (GW–FD).



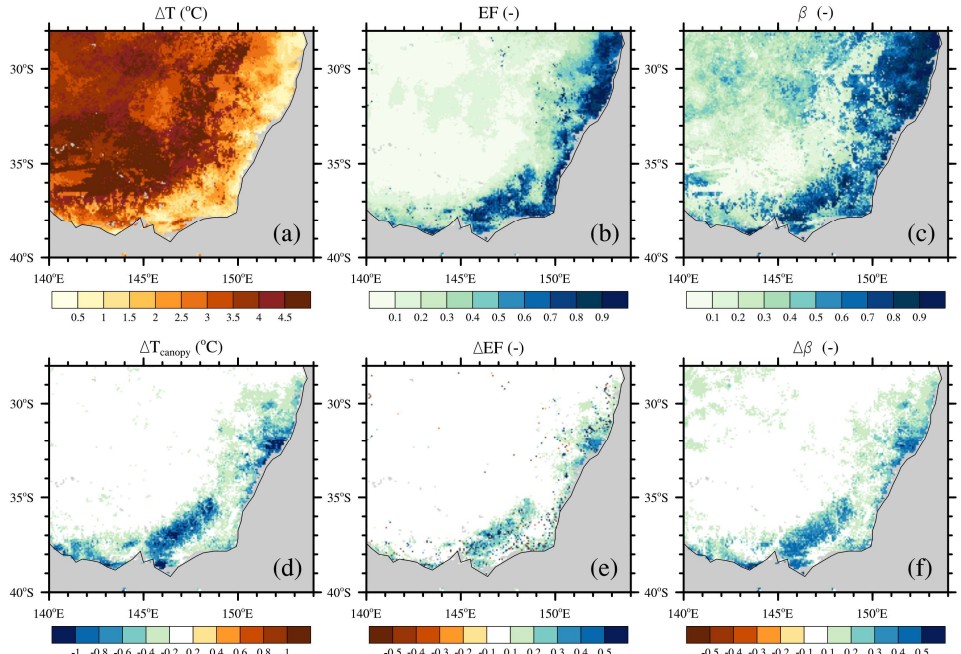

**Figure 4.** Land response to heatwaves during the recent drought. Panels (a)-(c) are the mean $T_{canopy}-T_{air}$ ($\Delta T$), evaporative fraction (EF), and soil water stress factor ($\beta$) in GW, respectively, during 2017−2019 summer heatwaves. Panel (d)-(f) are the difference (GW−FD) of $T_{canopy}$, EF and $\beta$. Note that the colour bar is switched between (d) and (e)-(f).

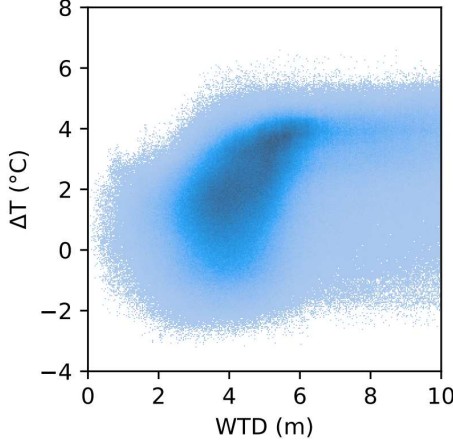

**Figure 5.** A density scatter plot of $T_{canopy}-T_{air}$ ($\Delta T$) versus water table depth (WTD) in GW simulations over forested areas in all heatwaves during 2000–2019. Every tree pixel on each heatwave day accounts for one record and the darker colours show higher recorded densities.

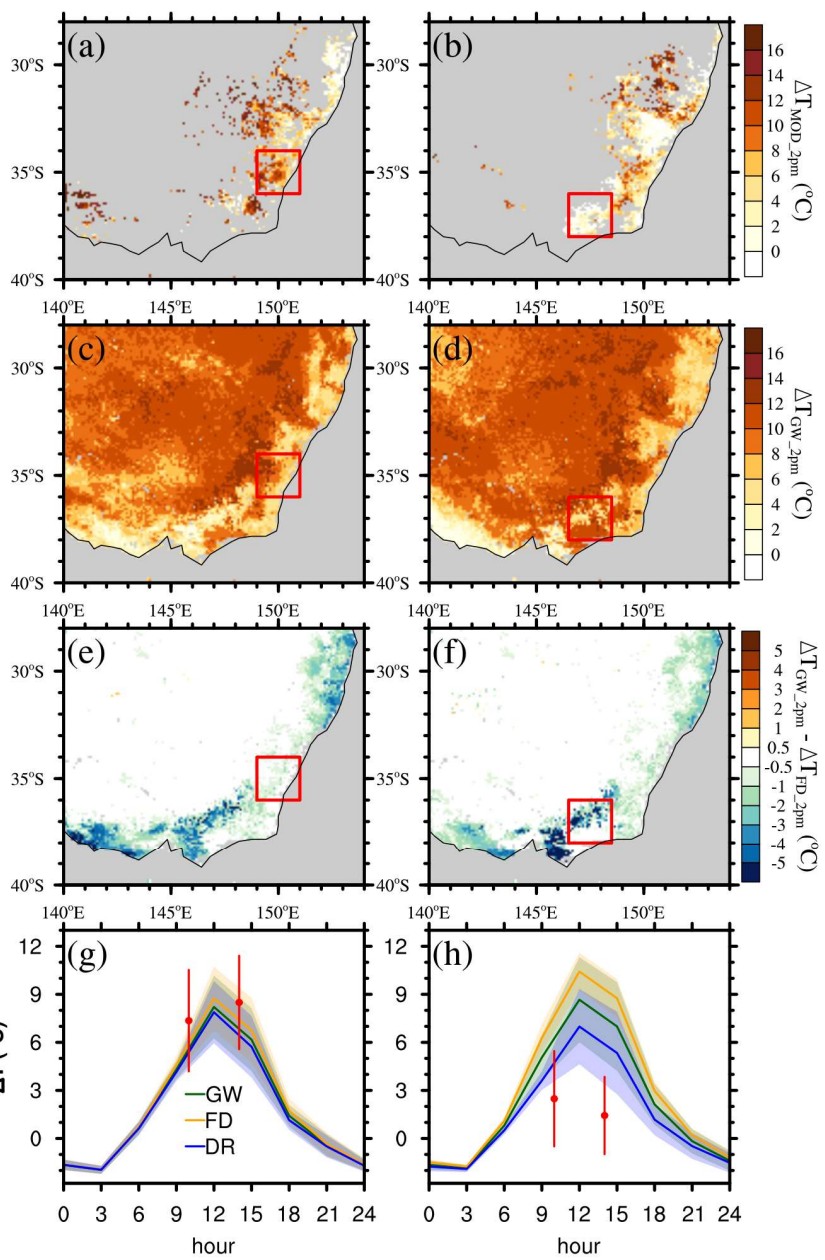

888

**Figure 6.** The simulation of two heatwaves on 15th (left column) and 25th January 2019 (right column). The first row shows the difference between MODIS land surface temperature (LST) and $T_{air}$ at 2 pm ($\Delta T_{MOD\_2pm}$) (only forested areas with good LST quality data are displayed). The second row is the GW simulation of $\Delta T$ at 2 pm ($\Delta T_{GW\_2pm}$). The third row is $\Delta T_{GW\_2pm}$ minus $\Delta T_{2pm}$ in FD ($\Delta T_{FD\_2pm}$). The last row is the diurnal cycle of $\Delta T$ over the selected regions shown by the red boxes in panels (e) and (f). In panel (e) and (f), the shadings show the uncertainty in every simulation defined as one standard deviation (SD) among the selected pixels. The red dots are MODIS LST minus $T_{air}$ with the uncertainty shown by the red error bars. For both regions, only pixels available in MODIS are shown.

895

896





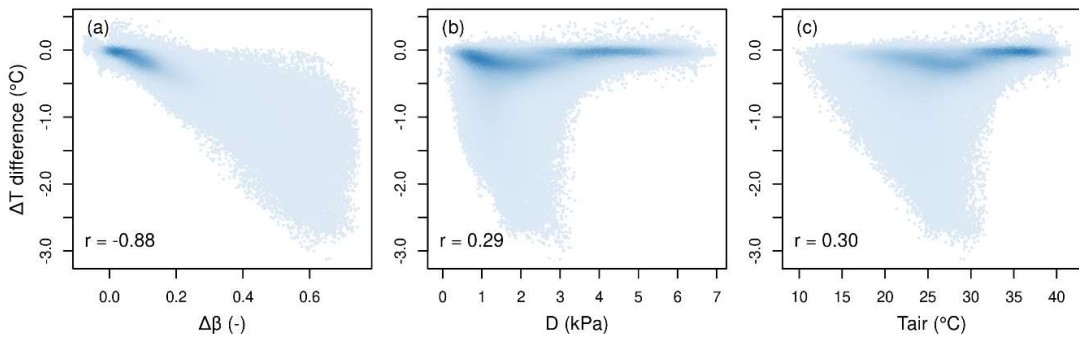

**Figure 7.** Density scatter plots showing the three factors that influence the difference in $T_{canopy}-T_{air}$ between GW and FD ($\Delta T$, expressed as GW−FD difference). (a) is $\Delta T$ difference against the $\beta$ difference (GW−FD) ($\Delta\beta$), (b) is $\Delta T$ difference against vapour pressure deficit (D), and (c) is $\Delta T$ difference against $T_{air}$. Each point corresponds to a tree pixel on a heatwave day in January 2019. The darker colours illustrate where the records are more dense. The correlation (r) between the x- and y-axis is shown in the bottom left of each panel.