# Peer review of "Exploring how groundwater buffers the influence of heatwaves on vegetation function during multi-year droughts"

_Earth System Dynamics, 2021_

## Referee Comment (RC3)

**Review of "Exploring how groundwater buffers the influence of heatwaves on vegetation function during multi-year droughts", by Mu et al.**

In this study, the authors analyze the influence on groundwater dynamics on land surface conditions during hot-dry compound events using dedicated land surface model simulations. The authors discover that groundwater can help maintain transpiration during the initial phase of a multi-year drought, and thereby dampen canopy temperatures during heatwave condition, but that this effect diminished beyond two years into the drought as the groundwater gets depleted.

The paper uses model simulations to assess an understudied process in land-atmosphere interactions, that is, groundwater-induced dampening of extreme heatwaves. The skill of the GW experiment regarding TWSA is quite impressive, especially since it appears that there has been no tuning. Moreover, the manuscript is well written, and the figures are generally clear. Also, the introduction reads very well.

This study thus overall demonstrates the potential to make a substantial contribution to the scientific literature. However, I have some concerns, which require minor revisions of the manuscript. In general, I could recommend publication of this study if the comments specified below are sufficiently addressed.

**General Comments**

1. My main concern relates to terminology and definitions: It appears that the paper is using inconsistent variable names, for instance in eq. 2 it uses $q_{re}$ for groundwater recharge while figure 2 uses Qrec for denoting apparently the same term. It is also not clear to me what is meant with 'vertical drainage' in fig. 2 (panel d): is this the vertical downward transport of water from the soil column to the aquifer to the soil column (what would normally be called groundwater recharge)? In that case 'recharge' represents the vertical upward transport of water from the aquifer to the soil column (as suggested in L270)? Overall, I'm confused by the terminology used in this context (recharge is normally used to denote the downward flux from soil to aquifer). Please carefully check throughout and make sure to use consistent and well-defined terms and variables throughout the manuscript and figures. Perhaps a schematic showing these fluxes across a vertical atmosphere+soil+aquifer profile could help as well (potentially with one panel per experiment).

**Specific comments**

1. L134: Also with the time-evolving meteorological forcing, right?

2.	L136: As we currently are in the CMIP6 era, I feel it could be more relevant to comment (also) on the status of groundwater modules in this generation of models.

3.	L209: In the case of water fluxes, a conservative remapping would be more appropriate. It's ok to leave it like this now, but please keep this in mind for future research.

4.	L123: From the context it appears that the simulations are run at 0.05° spatial resolution and 3h time step, but I suggest mentioning this somewhere explicitly in the method section, for instance in L124-127.

5.	Fig3a: the underscore in the right y-axis label can be omitted. How did you define forested area? All pixels in the model domain with 100% tree fraction? Please clarify in the caption.

**Textual comments**

1.	L248, caption figure 2 and elsewhere: replace '(total) evaporation' by 'evapotranspiration', whenever you are referring to the sum of transpiration and soil evaporation. Likewise, replace 'recharge' by 'groundwater recharge' if that is what you mean (though it looks like you mean something like 'soil moisture recharge' with this term, which appears odd to me).

2.	L346: 'estimated' > 'estimates'.

---

## Author Comment (AC1)

**Response to Reviewer 1**

**General comments**

This paper demonstrates the effect that modelling groundwater in the CABLE land-surface model has on droughts and heatwaves, using two droughts and multiple heatwaves in South East Australia as case studies. This is a very important and topical issue, and well within the ESD remit. The analysis is thorough and well-designed, and described in sufficient detail to allow reproduction. Particular attention is paid to understanding the mechanisms behind the results, which considerably strengthens the conclusions. The relevance to climate model projections is particularly well put, clearly stating the important implications of this study whilst carefully outlining uncertainties and avoiding over-generalisation.

All of the plots in both the main manuscript and the supplementary material are important for the arguments presented, and of a high production standard. The prose is well written, the structure is good, and there is a high attention to detail.

Overall, this is a very strong manuscript, which will make an important contribution to the field.

> We thank the reviewer for the positive summary of our work and we address the reviewer's concerns below. Comments are shown in black, with our response below in blue in each case.

**Specific comments**

Section 3.1: This section shows that CABLE-GW has a very good agreement with GRACE total water storage, and that the GW run has a better agreement with GRACE than the FD run. However, I'm not completely convinced by the conclusion that the underestimation of TWSA in the FD run is because of the lack of groundwater representation. (i.e. I don't think that other model deficiencies with the potential to reduce E have been ruled out). This is particularly the case as the accumulated P-E in GW run is still substantially different to GLEAM, showing that there is still an issue with the model even with groundwater included. To address this I would suggest (a) being a bit more cautious in the phrasing so that the text doesn't imply that including ground water is the only way to make the model match more closely to the observations and (b) including some text discussing possible reasons why the GW and GLEAM lines do not agree in figure 1b.

> We thank the reviewer for this comment, we fully agree that underestimation of TWSA is likely to also relate to other model biases. We will clarify in the manuscript that the underestimation of TWSA was in reference to FD compared to GW; these runs are identical for all parameterisations except groundwater, allowing us to attribute the bias to the groundwater processes:

"This underestimation in FD compared to GW is linked with the lack of aquifer water storage in the FD simulations which provides a reservoir of water that changes slowly and has a memory of previous wet/dry climate conditions (Figure 1a)."

With respect to the comment about GW and GLEAM lines not agreeing, we first note that GLEAM is itself a model (which ingested observations), so a mis-match does not necessarily point to a CABLE issue. To better illustrate this point we will add a second ET product, the Derived Optimal Linear Combination Evapotranspiration (DOLCE) (Hobeichi et al., 2021) estimate to Figure 1b (see blue line in the figure below). DOLCE is an observationally constrained product, similar to GLEAM. As can be seen, the GW simulation sits between the DOLCE and GLEAM estimates, whereas FD is outside the envelope of these observationally-constrained products.

[Figure]

Figure 1 panel b: accumulated P–E for the two droughts over S.E. Australia. It shows the accumulated P–E for two periods; the dark lines show the 2001–2009 Millennium drought (MD) and the light lines show the 2017-2019 recent drought (RD). The correlation (r) between the P and E is shown in the legend of panel (b).

We will add a sentence to address the reviewer's point about attributing all of the improvement in TWSA to GW alone, linking to both the GLEAM and DOLCE estimates, to highlight that we fully expect CABLE to still have other evaporative biases:

"Although the total land evaporation products display some differences, the GW simulations are closer overall to the DOLCE and GLEAM estimates. Whilst the biases in evapotranspiration can arise from multiple sources, the better match of GW than FD to the two observationally-constrained products implies that overall adding groundwater improves the simulations during droughts."

Line 229: "FD underestimates the magnitude of monthly TWSA variance (standard deviation, SD = 37.18 mm) compared to GRACE (47.74 mm) or GW (47.67 mm)": consider showing this explicitly in a plot, as it's an important result, which is difficult to read off Figure 1a.

We thank the reviewer for this point. We note that we did add these metrics to the top left corner of Figure 1a. To make this clearer to our readers, we will revise the colour of these metrics from black to blue in the figure panel.

Line 351: elaborate on why MODIS LST is lower than all the model lines in Figure 6h, even DR.

We checked the LAI in the underlying grid box in Figure 6h and it suggests a high LAI coverage, which would tend to imply that the MODIS LST is representing a good approximation of the canopy temperature. As a result, the lower MODIS LST$-$T$_{air}$ would tend to imply that CABLE is underestimating transpiration, leading to a greater T$_{canopy}$ $-$ T$_{air}$. We will make this clearer in the results:

"The shallower WTD region tends to have a high LAI coverage, implying that the MODIS LST likely represents a good approximation of the canopy temperature over this region. Consequently, the lower MODIS $\Delta$T implies that CABLE is likely underestimating transpiration, leading to an overestimation of $\Delta$T in all three simulations."

Line 369: "first ~two years of a multi-year drought": link this explicitly to plots (as far as I could see, this is the first time this was mentioned, but it's picked out as one of the main points of the study in both the abstract and the conclusion).

We will add shading similar to Figure 1 to Figure 3 to make it clearer where the drought periods are and hope this will make the link clearer. To clarify the statement of "first ~ two years of a multi-year drought", we will make this result clearer in the results and in the discussion:

"FD underestimates the magnitude of monthly TWSA variance (standard deviation, SD = 37.18 mm) compared to GRACE (47.74 mm) or GW (47.67 mm), particularly during the wetter periods (2000, 2011-2016) and the first ~2 years of the droughts (2001-2, 2017-8) (Figure 1a)",

"For all variables ($\Delta$T, EF, Et and $\beta$), the difference between GW and FD is greatest during the wetter periods (e.g. 2013) and in the first 1-2 years of the multi-year drought (2001-2002 for the Millennium Drought or 2017-2018 for the recent drought). After drought onset, the FD and GW simulations converge as depleting soil moisture reservoirs reduce the impact of groundwater on canopy cooling and evaporative fluxes",

and "We found that the influence of groundwater was the most important during the wetter periods and the first ~ two years of a multi-year drought (~2001-2002 and 2017-2018; Figure 1 and 3)".

Line 398-9 "Our regional based results support this hypothesis and in particular highlight the importance of groundwater for explaining the amplitude of fluxes in wet regions (Figure 1)" Elaborate on how Figure 1 shows this.

We thank the reviewer for spotting this mistake. We will change "wet regions" to "wetter periods" in the text:

"Our regional based results support this hypothesis and in particular highlight the importance of groundwater for explaining the amplitude of fluxes in wet periods, as well as sustaining evaporation during drought (Figure 1)".

We will also better highlight this behaviour during the wetter periods in the results:

"FD underestimates the magnitude of monthly TWSA variance (standard deviation, SD = 37.18 mm) compared to GRACE (47.74 mm) or GW (47.67 mm), particularly during the wetter periods (2000, 2011-2016) and the first ~2 years of the droughts (2001-2, 2017-8) (Figure 1a)",

and "For all variables ($\Delta$T, EF, Et and $\beta$), the difference between GW and FD is greatest during the wetter periods (e.g. 2013) and in the first 1-2 years of the multi-year drought (2001-2002 for the Millennium Drought or 2017-2018 for the recent drought)".

**Technical corrections**

Line 434-436: This sentence doesn't read well. Is it missing a "that" or an "and"?

To solve this comment, we will change the sentence to:

"Figure 1 gives us confidence that CABLE-GW is performing well, based on the evaluation against the GRACE, DOLCE and GLEAM products, and as well as previous work that showed the capacity of CABLE-GW to simulate E well (Decker et al., 2017). However, we also note that key model parameterisations that may influence the role of groundwater are particularly uncertain".

References:
Decker, M., Or, D., Pitman, A. and Ukkola, A.: New turbulent resistance parameterization for soil evaporation based on a pore-scale model: Impact on surface fluxes in CABLE, J. Adv. Model. Earth Syst., 9(1), 220–238, doi:10.1002/2016MS000832, 2017.
Hobeichi, S., Abramowitz, G. and Evans, J. P.: Robust historical evapotranspiration trends across climate regimes, Hydrol. Earth Syst. Sci., 25(7), 3855–3874, doi:10.5194/hess-25-3855-2021, 2021.

---

## Author Comment (AC2)

**Response to Reviewer 2**

In this study, Mu et al. evaluate the contribution of groundwater (GW) to vegetation water availability during heatwave and drought events in SE Australia. To do this, they implement a GW scheme in the CABLE land-surface model, and perform factorial simulations constrained by LAI to separate the contribution of GW to evapotranspiration and canopy temperature, which they compare with remote-sensing data.

The manuscript is concisely and clearly written, well structured and appropriately referenced and provides an important contribution to the land-surface modelling community. I find that two aspects could be improved:

> We thank the reviewer for the positive summary of our work and we address the reviewer's concerns below. Comments are shown in black, with our response below in blue in each case.

(i) After reading the title one would expect a greater focus on the impacts of GW on vegetation functional aspects (assimilation, stomatal conductance, transpiration, growth…), while the manuscript focuses mostly on hydrometeorology. Transpiration differences between the two simulations are only shown in Fig. 2, for 2019, but they could have been included in Fig. 3 and S6, to complement the discussion about the functional aspects. From the model simulations, one could additionally include, assimilation rates, stomatal conductance, NPP, etc.

> We thank the reviewer for these suggestions about other aspects of vegetation function not shown in the manuscript.

> We originally deployed this manuscript with the two considerations. First, as CABLE is run using prescribed LAI for these simulations, growth is not predictive (only gas exchange is predicted). This is a common approach used in a number of LSMs and is extremely helpful in these types of experimental setups as it isolates the change (i.e. directly to GW) without growth feedbacks. Second, our interest is on how GW sustains vegetation function via transpiration, which in turn may cool the boundary layer, when run in a coupled model environment. The carbon uptake plays no role in this link between the surface energy balance and the atmospheric feedback, hence we did not focus on it originally.

> Nevertheless, the reviewer's suggestions led us to reconsider these issues and we therefore will add spatial maps of GPP during the two droughts and the GPP difference between GW and FD. We will discuss the GPP result and explain why we do not focus on this in the rest of the manuscript.

> We thank the reviewer for the suggestion on transpiration plots. As requested, we will add transpiration plots to Figure 3 and a supplementary figure (see plots below).

[Figure]

Figure 3. Groundwater-induced differences in (a) $T_{canopy}$-$T_{air}$ ($\Delta T$), (b) evaporative fraction (EF), (c) transpiration (Et), and (d) water stress factor ($\beta$) during 2000-2019 summer heatwaves over forested areas. The left y-axis is the scale for boxes. The blue boxes refer to the GW experiment and the red boxes to FD. For each box, the middle line is the median, the upper border is the 75th percentile, and the lower border is 25th percentile. The right y-axis is the scale for the grey lines which display the difference in the medians (GW-FD). The shadings highlight the two drought periods.

[Figure]

Supplementary Figure: The difference of transpiration (Et) at 2pm between (a)-(b) GW and FD (EtGW_2pm-EtFD_2pm), and between (c)-(d) DR and GW (EtDR_2pm-EtGW_2pm). The left column is for 15th and the right is for 25th Jan 2019.

We will also add appropriate text:

"However, the strength of the cooling effect decreases as the droughts extends and the transpiration difference ($\Delta Et$, mm d$^{-1}$) diminishes quickly (Figure 3c) because the vegetation becomes increasingly water-stressed which consequently limits transpiration (Figure 3d)",

"While reductions of 5°C are clearly limited in spatial extent, the overall pattern of cooling is quite widespread, where is coincident with the groundwater-induced Et increase (indicated by the new supplementary figure), implying an extensive reduction in heat stress along shallow WTD coastal regions during heatwaves",

and "By enabling access to moisture in the deeper soil, the LSM simulates further cooling by 0.5–5°C across the forests associated with an Et increase of 25–250 W m$^{-2}$ (in the new supplementary figure)".

Moreover, the DR experiment is one of the most exciting aspects of the study since it highlights the relevance of the interactions between hydrology and physiology, but it is briefly discussed and shown only in Fig. 6. I find that a deeper analysis of the DR experiment and an additional figure on the impacts of the heatwaves for GW, FD and DR would increase the relevance of the study and better support the discussion around improvements to LSMs.

We are happy that the reviewer was interested in our DR simulations. We agree that this is a potentially important aspect of this work but note that the 2019 simulations were simply a proof-of-concept sensitivity experiment. Future work that allows roots to tap into the GW directly in LSMs, or that optimally set rooting depth by historical water availability, is a potential future avenue for research in LSMs (as noted in the discussion 4.3). However, as we lack the observations to meaningfully set root distributions across S.E. Australia, we do not plan to add additional DR plots (with the exception of a revision to Fig 6, see below). We will discuss this in the appropriate text:

"Given we lack the detailed observations to set root distributions across S.E. Australia, we undertake the DR experiment as a simple sensitivity study. We only run this experiment during January 2019, when the record-breaking heatwaves compound with the severe recent drought".

(ii) one of the key conclusions mentioned in the abstract and sections 4 and 5 is that GW helps sustaining higher transpiration rates in the first 1-2 years of multi-year droughts. However, most figures of the paper, and specifically the one showing transpiration differences, refer to the 2019 event. The figures showing differences between GW and FD for the full period do not separate specifically T from ET.

We thank the reviewer for this question. This paper aims to cover two aspects. First, it shows and explains the average behaviour of groundwater-induced transpiration on canopy

cooling during heat extremes, in the context of two major droughts. Second, we focus on the short-term feedback (days) during the 2019 summer heatwaves occurred at the end year of the recent multi-year drought. These answer our main questions from the perspective of two time scales. Figures 1-5 relate directly to the average behaviour across the two major droughts. The rest of the Figures concentrate on the Jan 2019 heatwave events during the recent drought. However, we acknowledge that we did not clearly indicate how GW allows transpiration to be sustained during the 1-2 years of the drought events in the context of these figures in the results. We will add the text below to the manuscript:

"However, the strength of the cooling effect decreases as the droughts extends and the transpiration difference ($\Delta$Et, mm d$^{-1}$) diminishes quickly (Figure 3c) because the vegetation becomes increasingly water-stressed which consequently limits transpiration (Figure 3d). For all variables ($\Delta$T, EF, Et and $\beta$), the difference between GW and FD is greatest during the wetter periods (e.g. 2013) and in the first 1-2 years of the multi-year drought (2001-2002 for the Millennium Drought or 2017-2018 for the recent drought). After drought onset, the FD and GW simulations converge as depleting soil moisture reservoirs reduce the impact of groundwater on canopy cooling and evaporative fluxes."

To address the comment on separating transpiration from total ET, we will add Figure 3c and the extra supplementary figure (see plots above) to substantiate our statement that groundwater sustains extra transpiration cooling canopy temperature.

Moreover, even though slightly bigger differences between GW and FD are seen in 2002 (1 year following drought onset) and 2017 (drought start), strong differences are found also in non-drought years, e.g. 2013 (Fig. 3 and S6). I do not think that the results, as currently shown, can support strong conclusions about the duration of the effect of GW on HW effects.

We think that our results do support conclusions about the duration of the effect of GW on water fluxes and canopy cooling (i.e. predominantly in the early years of drought). To better emphasise this point we will highlight the drought periods with shading in both Figures 1 and 3. However, the reviewer correctly points out that larger differences are seen during both non-drought periods and drought onset. We will make this clearer in the relevant result section:

"FD underestimates the magnitude of monthly TWSA variance (standard deviation, SD = 37.18 mm) compared to GRACE (47.74 mm) or GW (47.67 mm), particularly during the wetter periods (2000, 2011-2016) and the first ~2 years of the droughts (2001-2, 2017-8) (Figure 1a). For all variables ($\Delta$T, EF, Et and $\beta$), the difference between GW and FD is greatest during the wetter periods (e.g. 2013) and in the first 1-2 years of the multi-year drought (2001-2002 for the Millennium Drought or 2017-2018 for the recent drought)."

However, Figure 1a and Figure 3 (grey lines) show that as the droughts progress, the two experiments tend to converge in their canopy cooling and water fluxes and the impact of GW diminishes as soil water becomes increasingly limiting, supporting our conclusion of

a larger GW impact during drought onset. We will make this final point about the difference in canopy temperature in the manuscript to further emphasise the point:

"As the drought lengthens in time, the depletion of moisture gradually reduces this effect, from an average reduction of 0.52°C of the first 3 years to 0.16°C of the last 3 years in Millennium Drought (Figure 3a)",

"However, the strength of the cooling effect decreases as the droughts extends and the transpiration difference ($\Delta Et$, mm d$^{-1}$) diminishes quickly (Figure 3c) because the vegetation becomes increasingly water-stressed which consequently limits transpiration (Figure 3d). For all variables ($\Delta T$, EF, Et and $\beta$), the difference between GW and FD is greatest during the wetter periods (e.g. 2013) and in the first 1-2 years of the multi-year drought (2001-2002 for the Millennium Drought or 2017-2018 for the recent drought). After drought onset, the FD and GW simulations converge as depleting soil moisture reservoirs reduce the impact of groundwater on canopy cooling and evaporative fluxes",

and "We found that the influence of groundwater was the most important during the wetter periods and the first ~ two years of a multi-year drought (~2001-2002 and 2017-2018; Figure 1 and 3)".

Other comments:

L100: I think dimensional analysis gives the units of Fsoil as m3.m-3/s, or 1/s (to match the other two terms), can you confirm?

Thank you for spotting this, we will correct the units to 1/s.

L235: "much closer": indeed, but still very far.

Agreed, we will alter the text as,

"GW increases the evaporation relative to FD such that the accumulated P-E decreases from about 786 mm to 455 mm during the Millennium drought, which is within the range of DOLCE (460 mm) and GLEAM (97 mm) estimates",

and "Although the total land evaporation products display some differences, the GW simulations are closer overall to the DOLCE and GLEAM estimates".

L250: can be complemented by a map of root length in CABLE.

Thank you for this suggestion. We will add the below plot as the panel b to Figure S3 to show the root distribution among the simulated PFTs.

[Figure]

The root fraction (%) above a given depth (m)

.

L312-317: how much is this threshold dependent on model structure and parameterization? And how does it compare with the same results for the DR experiment?

The threshold of ~6 m does likely arise from the model assumption of a 4.6 m soil bucket which also sets the maximum rooting depth (roots are confined to the soil layers and do not extend to the GW aquifer in CABLE). The 4.6m depth comes from observational evidence of most roots being situated within the top 4.6m (Canadell et al. 1996). When the water table is below this depth, the water fluxes largely become uncoupled between the soil column and groundwater, leading to a diminished impact of GW below the ~6m threshold. The DR experiment uses the same soil depth assumption (it only differs from the GW simulation in having a larger fraction of roots situated in deep soil layers) and as such would display a similar threshold. As we only have DR outputs for January 2019, we did not explicitly quantify this.

However, clearly the threshold is CABLE-specific and we will acknowledge this in the manuscript:

"However, the absolute value of the threshold is likely CABLE-specific and associated with CABLE's assumption of a 4.6 m soil depth, which also sets the maximum rooting depth (roots can only extend to the bottom of the soil bucket and cannot access to the groundwater aquifer in CABLE). The CABLE soil depth comes from observational evidence of most roots being situated within the top 4.6 m (Canadell et al. 1996). Since the model assumes no roots in the groundwater aquifer, when the water table is below this depth, the water fluxes become largely uncoupled between the soil column and the groundwater aquifer, leading to a negligible impact of GW below ~6 m depth."

L325-331: very hard to compare panels a-b with c-d. Can you use a consistent mask?

If we used the same mask we would lose a lot of information from the model maps, which we feel is worth keeping. Instead, we have provided the masked plots in Figure S6 a-d and g-h, which compares the model simulated and MODIS-based ΔT.

L339-341: the label of Fig. 6e,f indicates "GW-FD". Can you check? I would find it important to show DR in more figures, as discussed above.

Thank you for the suggestion. We originally only provided "GW-FD" in Figure 6 but agree with the reviewer that the DR simulation is also of interest. As such we will show the difference "DR-GW" in Figure 6 g-h. The original panels g-h will be moved to a separate Figure 7.

L343: how can one compare panels a,b with g,h in Figure 6?

The panels g-h show the diurnal evolution of ΔT for the regions shown by the red boxes in panels a-f. As the maps only show behaviour at 2pm (when the afternoon MODIS overpass occurs), the line plots were created to show typical diurnal cycles. They allow the comparison of diurnal cycles across the three experiments (GW, FD and DR) in the context of the two available day-time MODIS overpasses.

L375-376: can you support this by separating results per WTD bins rather than simple visual inspection?

Thank you for this suggestion. We will add the metrics on the left bottom corner in each panel of Figure 4.

L376-377: Where can we see the time-dependence of this response?

We thank the reviewer to spot this point. We will clarify in the relative text:

"However, the strength of the cooling effect decreases as the droughts extends and the transpiration difference ($\Delta Et$, mm d$^{-1}$) diminishes quickly (Figure 3c) because the vegetation becomes increasingly water-stressed which consequently limits transpiration (Figure 3d). For all variables ($\Delta T$, EF, Et and $\beta$), the difference between GW and FD is greatest during the wetter periods (e.g. 2013) and in the first 1-2 years of the multi-year drought (2001-2002 for the Millennium Drought or 2017-2018 for the recent drought). After drought onset, the FD and GW simulations converge as depleting soil moisture reservoirs reduce the impact of groundwater on canopy cooling and evaporative fluxes",

and "Importantly, the role played by groundwater diminishes as the drought lengthens beyond two years (Figure 3)".

L408: specify what feedback is meant here

We mean the feedback from changes in the land surface fluxes on the boundary layer. We will change the text as "The lack of groundwater in many LSMs suggests a lack of this moderating process and consequently a risk of overestimating the positive feedback on the boundary layer in coupled climate simulations".

L448-449: rather than making a general statement, the authors could analyze variables related with the physiological responses (assimilation, stomatal conductance, WUE, NPP) to show that (if) GW matters.

As we explained above, we will also show GPP in the supplementary but concentrate on transpiration as it is the key variable for canopy temperatures and potential boundary layer feedbacks during heatwaves, which are the key research questions in this paper.

L486: what can we see a figure supporting this conclusion?

As we explained above, it comes from Figure 1 and Figure 3. We will highlight this finding in the results as per previous responses.

L487: the cooling effect is shown only for 2019, correct?

We saw a cooling effect during all 2001-2019 heatwave events, as well as the January 2019 heatwave. To make the point clearer, we will adjust the sentence as

"This cooled the forest canopy on average by 0.03–0.76 °C in heatwaves during 2001-2019 and as much as 5 °C in regions of shallow water table depths in the heatwave in January 2019, helping to moderate the heat stress on vegetation during heatwaves".

L490: this is not strongly supported by results (comments above)

Thanks. As per our replies to the previous comments, we will better explain this finding in the result section.

Reference:

Canadell, J., Jackson, R. B., Ehleringer, J. R., Mooney, H. A., Sala, O. E., and Schulze, E. D.: Maximum rooting depth of vegetation types at the global scale, Oecologia, 108(4), 583–595, 1996.

---

## Author Comment (AC3)

**Response to Reviewer 3**

**Review of "Exploring how groundwater buffers the influence of heatwaves on vegetation function during multi-year droughts", by Mu et al.**

In this study, the authors analyze the influence on groundwater dynamics on land surface conditions during hot-dry compound events using dedicated land surface model simulations. The authors discover that groundwater can help maintain transpiration during the initial phase of a multi-year drought, and thereby dampen canopy temperatures during heatwave condition, but that this effect diminished beyond two years into the drought as the groundwater gets depleted.

The paper uses model simulations to assess an understudied process in land-atmosphere interactions, that is, groundwater-induced dampening of extreme heatwaves. The skill of the GW experiment regarding TWSA is quite impressive, especially since it appears that there has been no tuning. Moreover, the manuscript is well written, and the figures are generally clear. Also, the introduction reads very well.

This study thus overall demonstrates the potential to make a substantial contribution to the scientific literature. However, I have some concerns, which require minor revisions of the manuscript. In general, I could recommend publication of this study if the comments specified below are sufficiently addressed.

> We thank the reviewer for the positive summary of our work and we address the reviewer's concerns below. Comments are shown in black, with our response below in blue in each case.

**General Comments**

1. My main concern relates to terminology and definitions: It appears that the paper is using inconsistent variable names, for instance in eq. 2 it uses qre for groundwater recharge while figure 2 uses Qrec for denoting apparently the same term. It is also not clear to me what is meant with 'vertical drainage' in fig. 2 (panel d): is this the vertical downward transport of water from the soil column to the aquifer to the soil column (what would normally be called groundwater recharge)? In that case 'recharge' represents the vertical upward transport of water from the aquifer to the soil column (as suggested in L270)? Overall, I'm confused by the terminology used in this context (recharge is normally used to denote the downward flux from soil to aquifer). Please carefully check throughout and make sure to use consistent and well-defined terms and variables throughout the manuscript and figures. Perhaps a schematic showing these fluxes across a vertical atmosphere+soil+aquifer profile could help as well (potentially with one panel per experiment).

We thank the reviewer for this comment. As we pointed out in Line 105-106, $q_{re}$ is the water flux between the aquifer and the bottom soil layer. The positive $q_{re}$ refers to the downward water flow from soil column to aquifer (i.e. vertical drainage, Dr), and the negative $q_{re}$ is the upward water movement from aquifer to soil column (i.e. recharge, Qrec). Recharge in our paper is not groundwater recharge but the recharge from the aquifer to the soil column. We will explicate this point in the manuscript,

"The positive $q_{re}$ refers to the downward water flow from soil column to aquifer (i.e. vertical drainage, Dr), and the negative $q_{re}$ is the upward water movement from aquifer to soil column (i.e. recharge, Qrec)".

To avoid the confusion, we will also clarify that 'recharge' in this paper is the recharge from aquifer to the soil column throughout the manuscript and figures.

We like the idea of a schematic and will endeavour to include this in a revised paper.

**Specific comments**

1. L134: Also with the time-evolving meteorological forcing, right?

Yes, the simulation from 1970-2019 is with time-evolving meteorological forcing. We will mention this point in the experiment design section.

2. L136: As we currently are in the CMIP6 era, I feel it could be more relevant to comment (also) on the status of groundwater modules in this generation of models.

We will try to find this information out for the CMIP 6 models and include a revised sentence in the revised manuscript if we can.

3. L209: In the case of water fluxes, a conservative remapping would be more appropriate. It's ok to leave it like this now, but please keep this in mind for future research.

We thank the reviewer for this suggestion, and we will explore this in future work.

4. L123: From the context it appears that the simulations are run at 0.05° spatial resolution and 3h time step, but I suggest mentioning this somewhere explicitly in the method section, for instance in L124-127.

Agree. We will edit this sentence as:

"To explore how groundwater influences droughts and heatwaves, we designed two experiments, with and without groundwater dynamics, driven by the same 3-hour meteorology forcing and land surface properties (see section 2.5 for datasets) for the period 1970-2019 at a 0.05° spatial resolution with a 3-hour time step."

5. Fig3a: the underscore in the right y-axis label can be omitted. How did you define forested area? All pixels in the model domain with 100% tree fraction? Please clarify in the caption.

Agree. We will remove the underscore in Figure 3.

In our simulations, there is only one vegetation type in each pixel, the dominant vegetation type (Figure S2a). For our study region, the forested area is the region is dominated by evergreen broadleaf forest (green area in Figure S2a). We will clarify the location in the caption of Figure 3:

"Groundwater-induced differences in (a) Tcanopy-Tair (ΔT), (b) evaporative fraction (EF), (c) transpiration (Et), and (d) water stress factor (β) during 2000-2019 summer heatwaves over forested areas (the green region in Figure S2a)".

**Textual comments**

1. L248, caption figure 2 and elsewhere: replace '(total) evaporation' by 'evapotranspiration', whenever you are referring to the sum of transpiration and soil evaporation. Likewise, replace 'recharge' by 'groundwater recharge' if that is what you mean (though it looks like you mean something like 'soil moisture recharge' with this term, which appears odd to me).

We thank the reviewer for this comment but we do not agree. Indeed, one of us was in a room when John Monteith complained bitterly about the word "evapotranspiration". The suggestion by the reviewer to add a schematic helps us clarify what "total evaporation" means which we hope will suffice but the term "evapotranspiration" may not be an appropriate way to express our fluxes.

As noted above, 'recharge' in our paper is not 'groundwater recharge' but 'recharge from the aquifer to the soil column'. We will clarify the recharge is the recharge from aquifer to soil column throughout the paper.

2. L346: 'estimated' > 'estimates'.

Thanks. This will be fixed as,

"Figure 7 shows the diurnal cycles of ΔT for the two selected regions (red boxes in Figure 6) compared with the MODIS LST estimates".

---

## Author Response (AR1)

**Response to Reviewer 1**

**General comments**

This paper demonstrates the effect that modelling groundwater in the CABLE land-surface model has on droughts and heatwaves, using two droughts and multiple heatwaves in South East Australia as case studies. This is a very important and topical issue, and well within the ESD remit. The analysis is thorough and well-designed, and described in sufficient detail to allow reproduction. Particular attention is paid to understanding the mechanisms behind the results, which considerably strengthens the conclusions. The relevance to climate model projections is particularly well put, clearly stating the important implications of this study whilst carefully outlining uncertainties and avoiding over-generalisation.

All of the plots in both the main manuscript and the supplementary material are important for the arguments presented, and of a high production standard. The prose is well written, the structure is good, and there is a high attention to detail.

Overall, this is a very strong manuscript, which will make an important contribution to the field.

We thank the reviewer for the positive summary of our work and we address the reviewer's concerns below. Comments are shown in black, with our response below in blue in each case.

**Specific comments**

Section 3.1: This section shows that CABLE-GW has a very good agreement with GRACE total water storage, and that the GW run has a better agreement with GRACE than the FD run. However, I'm not completely convinced by the conclusion that the underestimation of TWSA in the FD run is because of the lack of groundwater representation. (i.e. I don't think that other model deficiencies with the potential to reduce E have been ruled out). This is particularly the case as the accumulated P-E in GW run is still substantially different to GLEAM, showing that there is still an issue with the model even with groundwater included. To address this I would suggest (a) being a bit more cautious in the phrasing so that the text doesn't imply that including ground water is the only way to make the model match more closely to the observations and (b) including some text discussing possible reasons why the GW and GLEAM lines do not agree in figure 1b.

We thank the reviewer for this comment, we fully agree that underestimation of TWSA is likely to also relate to other model biases (see below). We have clarified in the manuscript that the underestimation of TWSA was in reference to FD compared to GW; these runs are identical for all parameterisations except groundwater, allowing us to attribute the bias to the groundwater processes:

Line 239-241, "This underestimation in FD compared to GW is linked with the lack of aquifer water storage in the FD simulations which provides a reservoir of water that changes slowly and has a memory of previous wet/dry climate conditions (Figure 1a)."

With respect to the comment about GW and GLEAM lines not agreeing, we first note that GLEAM is itself a model (which ingested observations), so a mis-match does not necessarily point to a CABLE issue. To better illustrate this point we have added a second ET product, the Derived Optimal Linear Combination Evapotranspiration (DOLCE) (Hobeichi et al., 2021) estimate to Figure 1b (see blue line in the figure below). DOLCE is an observationally constrained product, similar to GLEAM. As can be seen, the GW simulation sits between the DOLCE and GLEAM estimates, whereas FD is outside the envelope of these observationally-constrained products.

[Figure]

Figure 1 panel b: accumulated P–E for the two droughts over S.E. Australia. It shows the accumulated P–E for two periods; the dark lines show the 2001–2009 Millennium drought (MD) and the light lines show the 2017-2019 recent drought (RD). The correlation (r) between the P and E is shown in the legend of panel (b).

We now add a sentence to address the reviewer's point about attributing all of the improvement in TWSA to GW alone, linking to both the GLEAM and DOLCE estimates, to highlight that we fully expect CABLE to still have other evaporative biases:

Line 248-252, "Although the evapotranspiration products display some differences, the GW simulations are closer overall to both the DOLCE and the GLEAM, observational-constrained estimates. The better match of GW than FD to the two evapotranspiration products implies that adding groundwater improves the simulations during droughts, whilst the remaining mismatch would tend to suggest further biases in simulated evapotranspiration arising from multiple sources (e.g., a mis-match in leaf area index, or contributions from the understorey)".

Line 229: "FD underestimates the magnitude of monthly TWSA variance (standard deviation, SD = 37.18 mm) compared to GRACE (47.74 mm) or GW (47.67 mm)": consider showing this explicitly in a plot, as it's an important result, which is difficult to read off Figure 1a.

We thank the reviewer for this point. We note that we did add these metrics to the top left corner of Figure 1a. To make this clearer to our readers, we have revised the colour of these metrics from black to blue in the figure panel.

Line 351: elaborate on why MODIS LST is lower than all the model lines in Figure 6h, even DR.

We checked the LAI in the underlying grid box in Figure 6h (current Figure 7b) and it suggests a high LAI coverage, which would tend to imply that the MODIS LST is representing a good approximation of the canopy temperature. As a result, the lower MODIS LST−$T_{air}$ would imply that CABLE is underestimating transpiration, leading to a greater $T_{canopy} - T_{air}$. We now make this clearer in the results:

Line 380-383, "The shallower WTD region (Figure 7b) tends to have a high LAI coverage, implying that the MODIS LST represents a good approximation of the canopy temperature over this region. Consequently, the lower MODIS $\Delta T$ implies that CABLE is likely underestimating transpiration, leading to an overestimation of $\Delta T$ in all three simulations."

Line 369: "first ~two years of a multi-year drought": link this explicitly to plots (as far as I could see, this is the first time this was mentioned, but it's picked out as one of the main points of the study in both the abstract and the conclusion).

We have added shading similar to Figure 1 to Figure 3 to make it clearer where the drought periods are and hope this will make the link clearer. To clarify the statement of "first ~ two years of a multi-year drought", we now make this result clearer in both the result and the discussion sections:

Line 236-239, "FD underestimates the magnitude of monthly TWSA variance (standard deviation, SD = 37.18 mm) compared to GRACE (47.74 mm) or GW (47.67 mm), particularly during the wetter periods (2000, 2011-2016) and the first ~2 years of the droughts (2001-2, 2017-8) (Figure 1a)",

Line 319-323, "For all variables ($\Delta T$, EF, Et and $\beta$), the difference between GW and FD is greatest during the wetter periods (e.g., 2013) and in the first 1–2 years of the multi-year drought (2001–2002 for the Millennium Drought or 2017–2018 for the recent drought). After drought becomes well established, the FD and GW simulations converge as depleting soil moisture reservoirs reduce the impact of groundwater on canopy cooling and evaporative fluxes",

and Line 399-401, "We found that the influence of groundwater was most important during the wetter periods and the first ~ two years of a multi-year drought (~2001–2002 and 2017–2018; Figure 1 and 3)".

Line 398-9 "Our regional based results support this hypothesis and in particular highlight the importance of groundwater for explaining the amplitude of fluxes in wet regions (Figure 1)" Elaborate on how Figure 1 shows this.

We thank the reviewer for spotting this mistake. We have changed "wet regions" to "wetter periods" in the text:

Line 432-434, "Our regional based results support this hypothesis and in particular highlight the importance of groundwater for explaining the amplitude of fluxes in wet periods, as well as sustaining evapotranspiration during drought (Figure 1)".

We also better highlight this behaviour during the wetter periods in the results:

Line 236-239, "FD underestimates the magnitude of monthly TWSA variance (standard deviation, SD = 37.18 mm) compared to GRACE (47.74 mm) or GW (47.67 mm), particularly during the wetter periods (2000, 2011-2016) and the first ~2 years of the droughts (2001-2, 2017-8) (Figure 1a)",

and Line 319-321, "For all variables ($\Delta T$, EF, Et and $\beta$), the difference between GW and FD is greatest during the wetter periods (e.g., 2013) and in the first 1–2 years of the multi-year drought (2001–2002 for the Millennium Drought or 2017–2018 for the recent drought)".

**Technical corrections**

Line 434-436: This sentence doesn't read well. Is it missing a "that" or an "and"?

Thanks. To solve this comment, we have changed the sentence to:

Line 468-471, "Figure 1 gives us confidence that CABLE-GW is performing well, based on the evaluation against the GRACE, DOLCE and GLEAM products, as well as previous work that showed the capacity of CABLE-GW to simulate E well (Decker, 2015; Decker et al., 2017). However, we also note that key model parameterisations that may influence the role of groundwater are particularly uncertain".

References:
Hobeichi, S., Abramowitz, G. and Evans, J. P.: Robust historical evapotranspiration trends across climate regimes, Hydrol. Earth Syst. Sci., 25(7), 3855–3874, doi:10.5194/hess-25-3855-2021, 2021.

**Response to Reviewer 2**

In this study, Mu et al. evaluate the contribution of groundwater (GW) to vegetation water availability during heatwave and drought events in SE Australia. To do this, they implement a GW scheme in the CABLE land-surface model, and perform factorial simulations constrained by LAI to separate the contribution of GW to evapotranspiration and canopy temperature, which they compare with remote-sensing data.

The manuscript is concisely and clearly written, well structured and appropriately referenced and provides an important contribution to the land-surface modelling community. I find that two aspects could be improved:

We thank the reviewer for the positive summary of our work and we address the reviewer's concerns below. Comments are shown in black, with our response below in blue in each case.

(i) After reading the title one would expect a greater focus on the impacts of GW on vegetation functional aspects (assimilation, stomatal conductance, transpiration, growth…), while the manuscript focuses mostly on hydrometeorology. Transpiration differences between the two simulations are only shown in Fig. 2, for 2019, but they could have been included in Fig. 3 and S6, to complement the discussion about the functional aspects. From the model simulations, one could additionally include, assimilation rates, stomatal conductance, NPP, etc.

We thank the reviewer for these suggestions about other aspects of vegetation function not shown in the manuscript.

We originally wrote this manuscript with the two considerations. First, as CABLE is run using prescribed LAI for these simulations, there is no growth (only gas exchange–is predicted, e.g. photosynthesis and stomatal conductance.). This is a common approach used in a number of LSMs and is extremely helpful in these types of experimental setups as it isolates the change (i.e., directly to GW) without growth feedbacks (which would alter leaf area and so, evaporation). Second, our interest is specifically: how does (via what mechanisms) GW sustains vegetation function via transpiration (i.e., is the change large enough to cool the boundary layer, which would in turn, impact coupled modelled simulations)?

Nevertheless, the reviewer's suggestions led us to reconsider these issues and we therefore added spatial maps of GPP during the two droughts and the GPP difference between GW and FD (Figure S4). We now also discuss the GPP result and explain why we do not focus on this in the rest of the manuscript in Line 256-261:

"Plant photosynthesis assimilation rates are associated with transpiration via stomata conductance. Figure S4 presents the spatial maps of gross primary productivity (GPP) during the two droughts. GW simulations increase carbon uptake by 50~300 g C yr$^{-1}$, particularly along the coasts (Figure S4c,f). However, since CABLE uses a prescribed LAI and does not simulate any feedback between water availability and plant growth (e.g., defoliation) and its impact on GPP, we only focus on how GW influences evapotranspiration and the surface energy balance in the subsequent sections."

We thank the reviewer for the suggestion on transpiration plots. As requested, we have added transpiration plots to Figure 3 and Figure S8 (see plots below).

[Figure]

Figure 3. Groundwater-induced differences in (a) $T_{canopy}$-$T_{air}$ ($\Delta T$), (b) evaporative fraction (EF), (c) transpiration (Et), and (d) water stress factor ($\beta$) during 2000-2019 summer heatwaves over forested areas. The left y-axis is the scale for boxes. The blue boxes refer to the GW experiment and the red boxes to FD. For each box, the middle line is the median, the upper border is the 75th percentile, and the lower border is 25th percentile. The right y-axis is the scale for the grey lines which display the difference in the medians (GW-FD). The shadings highlight the two drought periods.

[Figure]

Figure S8. The difference of transpiration (Et) at 2pm between (a)-(b) GW and FD ($Et_{GW\_2pm}$-$Et_{FD\_2pm}$), and between (c)-(d) DR and GW ($Et_{DR\_2pm}$-$Et_{GW\_2pm}$). The left column is for 15th and the right is for 25th Jan 2019.

We have also edited the appropriate text:

Line 317-319, "However, the strength of the cooling effect decreases as the droughts extends and the transpiration difference ($\Delta$Et, mm d$^{-1}$) diminishes quickly (Figure 3c) because the vegetation becomes increasingly water-stressed (Figure 3d) which consequently limits transpiration",

Line 362-364, "While reductions of 5ºC are clearly limited in spatial extent, the overall pattern of cooling is quite widespread, and coincident with the groundwater-induced Et increase (Figure S8a-b), implying a reduction in heat stress along coastal regions with a shallow WTD during heatwaves",

and Line 370-371, "By enabling access to moisture in the deeper soil, the LSM simulates further cooling by 0.5–5°C across the forests associated with an Et increase of 25–250 W m$^{-2}$ (Figure S8c-d)".

Moreover, the DR experiment is one of the most exciting aspects of the study since it highlights the relevance of the interactions between hydrology and physiology, but it is briefly discussed and shown only in Fig. 6. I find that a deeper analysis of the DR experiment and an additional figure on the impacts of the heatwaves for GW, FD and DR would increase the relevance of the study and better support the discussion around improvements to LSMs.

We are happy that the reviewer was interested in our DR simulations. We agree that this is a potentially important aspect of this work but note that the 2019 simulations were simply a proof-of-concept sensitivity experiment. Future work that allows roots to tap into the GW directly in LSMs, or that optimally set rooting depth by historical water availability, is a potential future avenue for research in LSMs (as noted in the discussion 4.3). However, as we lack the observations to meaningfully set root distributions across S.E. Australia, we do not plan to add additional DR plots (with the exception of a revision to Fig 6, see below). We now discuss this in Line 178-180:

"Given we lack the detailed observations to set root distributions across S.E. Australia, we undertake the DR experiment as a simple sensitivity study. We only run this experiment during January 2019, when the record-breaking heatwaves compound with the severe recent drought".

(ii) one of the key conclusions mentioned in the abstract and sections 4 and 5 is that GW helps sustaining higher transpiration rates in the first 1-2 years of multi-year droughts. However, most figures of the paper, and specifically the one showing transpiration differences, refer to the 2019 event. The figures showing differences between GW and FD for the full period do not separate specifically T from ET.

We thank the reviewer for this question. This paper aims to cover two aspects. First, it shows and explains the average behaviour of groundwater-induced transpiration on canopy cooling during heat extremes, in the context of two major droughts. Second, we focus on the short-term feedback (days) during the 2019 summer heatwaves occurred at the end year of the recent multi-year drought. These answer our main questions from the perspective of two time scales. Figures 1-5 relate directly to the average behaviour across the two major droughts. The rest of the Figures concentrate on the Jan 2019 heatwave events during the recent drought. However, we acknowledge that we did not clearly indicate how GW allows transpiration to be sustained during the 1-2 years of the drought events in the context of these figures in the results. We now have added the text below to the manuscript:

Line 317-323, "However, the strength of the cooling effect decreases as the droughts extends and the transpiration difference ($\Delta Et$, mm d$^{-1}$) diminishes quickly (Figure 3c) because the vegetation becomes increasingly water-stressed (Figure 3d) which consequently limits transpiration. For all variables ($\Delta T$, EF, Et and $\beta$), the difference between GW and FD is greatest during the wetter periods (e.g., 2013) and in the first 1–2 years of the multi-year drought (2001–2002 for the Millennium Drought or 2017–2018 for the recent drought). After the drought becomes well established, the FD and GW simulations converge as depleting soil moisture reservoirs reduce the impact of groundwater on canopy cooling and evaporative fluxes."

To address the comment on separating transpiration from total ET, we have added Figure 3c and the Figure S8 (see plots above) to substantiate our statement that groundwater sustains extra transpiration cooling canopy temperature.

Moreover, even though slightly bigger differences between GW and FD are seen in 2002 (1 year following drought onset) and 2017 (drought start), strong differences are found also in non-drought years, e.g. 2013 (Fig. 3 and S6). I do not think that the results, as currently shown, can support strong conclusions about the duration of the effect of GW on HW effects.

We think that our results do support conclusions about the duration of the effect of GW on water fluxes and canopy cooling (i.e., predominantly in the early years of drought). To better emphasise this point we have highlighted the drought periods with shading in both Figures 1 and 3. However, the reviewer correctly points out that larger differences are seen during both non-drought periods and drought onset. We now make this clearer in the text:

Line 236-239, "FD underestimates the magnitude of monthly TWSA variance (standard deviation, SD = 37.18 mm) compared to GRACE (47.74 mm) or GW (47.67 mm), particularly during the wetter periods (2000, 2011-2016) and the first ~2 years of the droughts (2001-2, 2017-8) (Figure 1a)",

and Line 319-321, "For all variables ($\Delta T$, EF, Et and $\beta$), the difference between GW and FD is greatest during the wetter periods (e.g., 2013) and in the first 1–2 years of the multi-year drought (2001–2002 for the Millennium Drought or 2017–2018 for the recent drought)".

However, Figure 1a and Figure 3 (grey lines) show that as the droughts progress, the two experiments tend to converge in their canopy cooling and water fluxes and the impact of GW diminishes as soil water becomes increasingly limiting, supporting our conclusion of a larger GW impact during drought onset. We now clarify this final point about the difference in canopy temperature in the manuscript:

Line 313-314, "As the drought lengthens in time, the depletion of moisture gradually reduces this effect, from an average reduction of 0.52°C of the first 3 years to 0.16°C of the last 3 years in Millennium Drought (Figure 3a)",

Line 317-323, "However, the strength of the cooling effect decreases as the droughts extends and the transpiration difference ($\Delta Et$, mm d$^{-1}$) diminishes quickly (Figure 3c) because the vegetation becomes increasingly water-stressed (Figure 3d) which consequently limits transpiration. For all variables ($\Delta T$, EF, Et and $\beta$), the difference between GW and FD is greatest during the wetter periods (e.g., 2013) and in the first 1–2 years of the multi-year drought (2001–2002 for the Millennium Drought or 2017–

2018 for the recent drought). After the drought becomes well established, the FD and GW simulations converge as depleting soil moisture reservoirs reduce the impact of groundwater on canopy cooling and evaporative fluxes",

and Line 399-401, "We found that the influence of groundwater was most important during the wetter periods and the first ~ two years of a multi-year drought (~2001–2002 and 2017–2018; Figure 1 and 3)".

Other comments:

L100: I think dimensional analysis gives the units of Fsoil as m3.m-3/s, or 1/s (to match the other two terms), can you confirm?

Thank you for spotting this, we have corrected the units to s$^{-1}$. We note that the units for Fsoil are incorrect in the papers we cite, but we have confirmed the units are indeed s$^{-1}$.

L235: "much closer": indeed, but still very far.

Agreed, we altered the text as described below:

Line 242-244, "GW increases the evapotranspiration relative to FD such that the accumulated P-E decreases from about 786 mm to 455 mm during the Millennium drought, which is within the range of DOLCE (460 mm) and GLEAM (97 mm) estimates",

and Line 248-249, "Although the evapotranspiration products display some differences, the GW simulations are closer overall to both the DOLCE and the GLEAM, observational-constrained estimates".

L250: can be complemented by a map of root length in CABLE.

We have added the below plot as the panel b to Figure S3 (currently Figure S2) to show the root distribution among the simulated PFTs.

[Figure]

Figure S2b The root fraction (%) above a given depth (m)

L312-317: how much is this threshold dependent on model structure and parameterization? And how does it compare with the same results for the DR experiment?

The threshold of ~6 m does likely arise in part from the model assumption of a 4.6 m soil bucket which also sets the maximum rooting depth (roots are confined to the soil layers and do not extend to the GW aquifer in CABLE). The 4.6 m depth comes from observational evidence of most roots being situated within the top 4.6 m (Canadell et al. 1996). When the water table is below this depth, the water fluxes largely become uncoupled between the soil column and groundwater, leading to a diminished impact of GW below the ~6 m threshold. The DR experiment uses the same soil depth assumption (it only differs from the GW simulation in having a larger fraction of roots situated in deep soil layers) and as such would display a similar threshold. As we only have DR outputs for January 2019, we did not explicitly quantify this.

However, clearly the threshold is CABLE-specific and we have acknowledged this in the manuscript:

Line 340-345, "However, the absolute value of the threshold is likely CABLE-specific and associated with the assumption of a 4.6 m soil depth, which also sets the maximum rooting depth (roots can only extend to the bottom of the soil and cannot directly access the groundwater aquifer in CABLE). The CABLE soil depth comes from observational evidence of most roots being situated within the top 4.6 m (Canadell et al. 1996). Since the model assumes no roots exist in the groundwater aquifer, when the water table is below this depth, the water fluxes become largely uncoupled between the soil column and the groundwater aquifer, leading to a negligible impact of GW below ~6 m depth."

L325-331: very hard to compare panels a-b with c-d. Can you use a consistent mask?

If we used the same mask we would lose a lot of information from the model maps, which we feel is worth keeping. Instead, we have provided the masked plots in Figure S6 a-d and g-h (currently Figure S7), which compares the model simulated and MODIS-based ΔT.

L339-341: the label of Fig. 6e,f indicates "GW-FD". Can you check? I would find it important to show DR in more figures, as discussed above.

We originally only provided "GW-FD" in Figure 6 but agree with the reviewer that the DR simulation is also of interest. As such we now show the difference "DR-GW" in Figure 6 g-h. The original panels g-h have been moved to a separate Figure 7.

L343: how can one compare panels a,b with g,h in Figure 6?

The panels g-h show the diurnal evolution of ΔT for the regions shown by the red boxes in panels a-f. As the maps only show behaviour at 2 pm (when the afternoon MODIS overpass occurs), the line plots were created to show typical diurnal cycles. They allow the comparison of diurnal cycles across the three experiments (GW, FD and DR) in the context of the two available day-time MODIS overpasses.

L375-376: can you support this by separating results per WTD bins rather than simple visual inspection?

Thank you for this suggestion. We now add the metrics on the left bottom corner in each panel of Figure 4.

Please note that we accidentally plotted the Millennium drought instead of the recent drought in Figure 4. We have now corrected the mistake, with the fixed plot supporting our original conclusions.

L376-377: Where can we see the time-dependence of this response?

We agree our original text was not clear. We have therefore clarified in the text:

Line 317-323, "However, the strength of the cooling effect decreases as the droughts extends and the transpiration difference ($\Delta$Et, mm d$^{-1}$) diminishes quickly (Figure 3c) because the vegetation becomes increasingly water-stressed (Figure 3d) which consequently limits transpiration. For all variables ($\Delta$T, EF, Et and $\beta$), the difference between GW and FD is greatest during the wetter periods (e.g., 2013) and in the first 1–2 years of the multi-year drought (2001–2002 for the Millennium Drought or 2017–2018 for the recent drought). After the drought becomes well established, the FD and GW simulations converge as depleting soil moisture reservoirs reduce the impact of groundwater on canopy cooling and evaporative fluxes".

and Line 409-410, "Importantly, the role played by groundwater diminishes as the drought lengthens beyond two years (Figure 3)".

L408: specify what feedback is meant here

We mean the feedback from changes in the land surface fluxes on the boundary layer. We have explicated this point in Line 441-443, "The lack of groundwater in many LSMs suggests a lack of this moderating process and consequently a risk of overestimating the positive feedback on the boundary layer in coupled climate simulations".

L448-449: rather than making a general statement, the authors could analyze variables related with the physiological responses (assimilation, stomatal conductance, WUE, NPP) to show that (if) GW matters.

As we explained above, we now also show GPP in Figure S4 but concentrate on transpiration as it is the key variable for canopy temperatures and potential boundary layer feedbacks during heatwaves, which are the key research questions in this paper.

L486: what can we see a figure supporting this conclusion?

As we explained above, it comes from Figure 1 and Figure 3. We have highlighted this finding in the results as per previous responses.

L487: the cooling effect is shown only for 2019, correct?

We saw a cooling effect during all 2001-2019 heatwave events, as well as the January 2019 heatwave. Importantly, in our analysis of 2001-2019, we are shown the daily average of summer heatwave days

across years, whereas in 2019, we are examining diurnal differences (this explains the differences in the magnitudes of the temperature change). To make the point clearer, we now adjust the sentence as

Line 522-524, "This cooled the forest canopy on average by 0.03–0.76°C in heatwaves during 2001–2019 and as much as 5°C in regions of shallow water table depths in the heatwave in January 2019, helping to moderate the heat stress on vegetation during heatwaves".

L490: this is not strongly supported by results (comments above)

Thanks. As per our replies to the previous comments, we have better explained this finding in the result section.

Reference:
Canadell, J., Jackson, R. B., Ehleringer, J. R., Mooney, H. A., Sala, O. E., and Schulze, E. D.: Maximum rooting depth of vegetation types at the global scale, Oecologia, 108(4), 583–595, 1996.

**Response to Reviewer 3**

**Review of "Exploring how groundwater buffers the influence of heatwaves on vegetation function during multi-year droughts", by Mu et al.**

In this study, the authors analyze the influence on groundwater dynamics on land surface conditions during hot-dry compound events using dedicated land surface model simulations. The authors discover that groundwater can help maintain transpiration during the initial phase of a multi-year drought, and thereby dampen canopy temperatures during heatwave condition, but that this effect diminished beyond two years into the drought as the groundwater gets depleted.

The paper uses model simulations to assess an understudied process in land-atmosphere interactions, that is, groundwater-induced dampening of extreme heatwaves. The skill of the GW experiment regarding TWSA is quite impressive, especially since it appears that there has been no tuning. Moreover, the manuscript is well written, and the figures are generally clear. Also, the introduction reads very well.

This study thus overall demonstrates the potential to make a substantial contribution to the scientific literature. However, I have some concerns, which require minor revisions of the manuscript. In general, I could recommend publication of this study if the comments specified below are sufficiently addressed.

We thank the reviewer for the positive summary of our work and we address the reviewer's concerns below. Comments are shown in black, with our response below in blue in each case.

**General Comments**

1. My main concern relates to terminology and definitions: It appears that the paper is using inconsistent variable names, for instance in eq. 2 it uses qre for groundwater recharge while figure 2 uses Qrec for denoting apparently the same term. It is also not clear to me what is meant with 'vertical drainage' in fig. 2 (panel d): is this the vertical downward transport of water from the soil column to the aquifer to the soil column (what would normally be called groundwater recharge)? In that case 'recharge' represents the vertical upward transport of water from the aquifer to the soil column (as suggested in L270)? Overall, I'm confused by the terminology used in this context (recharge is normally used to denote the downward flux from soil to aquifer). Please carefully check throughout and make sure to use consistent and well-defined terms and variables throughout the manuscript and figures. Perhaps a schematic showing these fluxes across a vertical atmosphere+soil+aquifer profile could help as well (potentially with one panel per experiment).

We thank the reviewer for this comment. As we pointed out in Line 105-106, $q_{re}$ is the water flux between the aquifer and the bottom soil layer. The positive $q_{re}$ refers to the downward water flow from soil column to aquifer (i.e., vertical drainage, Dr), and the negative $q_{re}$ is the upward water movement from aquifer to soil column (i.e., recharge, Qrec). Recharge in our paper is not groundwater recharge but the recharge from the aquifer to the soil column. We now explain this on Line 112-114, "The positive $q_{re}$ refers to the downward water flow from soil column to aquifer (i.e. vertical drainage, Dr), and the negative $q_{re}$ is the upward water movement from aquifer to soil column (i.e.

recharge, Qrec)".

To avoid the confusion, we have also clarified that 'recharge' in this paper is the recharge from aquifer to the soil column throughout the manuscript and figures.

We considered the suggestion on including a schematic carefully. On reflection we think we have clarified the terms and definitions of 'recharge' throughout the manuscript by editing the text. We think we have done this clearly enough to negate the need for a schematic.

**Specific comments**

1. L134: Also with the time-evolving meteorological forcing, right?

Yes, the simulation from 1970-2019 is with time-evolving meteorological forcing. We have added this point in Line 126-128, "To explore how groundwater influences droughts and heatwaves, we designed two experiments, with and without groundwater dynamics, driven by the same 3-hour time-evolving meteorology forcing and the same land surface properties (see section 2.5 for datasets) for the period 1970-2019 at a 0.05° spatial resolution with a 3-hour time step".

2. L136: As we currently are in the CMIP6 era, I feel it could be more relevant to comment (also) on the status of groundwater modules in this generation of models.

We agree with the reviewer that ideally we could discuss the status of groundwater models in CMIP6 models. However, we found this information is not (yet) readily available for CMIP6 models and it would require a substantial effort to establish which models consider groundwater dynamics in the many (now more than 100) CMIP6 models available. We also do not expect major differences in the representation of groundwater between CMIP5 and CMIP6 models, and as such we have opted to keep the sentence as is.

3. L209: In the case of water fluxes, a conservative remapping would be more appropriate. It's ok to leave it like this now, but please keep this in mind for future research.

We thank the reviewer for this suggestion, and we will explore this in future work.

4. L123: From the context it appears that the simulations are run at 0.05° spatial resolution and 3h time step, but I suggest mentioning this somewhere explicitly in the method section, for instance in L124-127.

Agree. We have edited this sentence as:

Line 126-128, "To explore how groundwater influences droughts and heatwaves, we designed two experiments, with and without groundwater dynamics, driven by the same 3-hour time-evolving meteorology forcing and the same land surface properties (see section 2.5 for datasets) for the period 1970-2019 at a 0.05° spatial resolution with a 3-hour time step".

5. Fig3a: the underscore in the right y-axis label can be omitted. How did you define forested area? All pixels in the model domain with 100% tree fraction? Please clarify in the caption.

We have removed the underscore in Figure 3.

In our simulations, there is only one vegetation type in each pixel, the dominant vegetation type (Figure S2a). For our study region, the forested area is the region is dominated by evergreen broadleaf forest (green area in Figure S2a). We have clarified the location in the caption of Figure 3:

"Groundwater-induced differences in (a) $T_{canopy}$-$T_{air}$ ($\Delta T$), (b) evaporative fraction (EF), (c) transpiration (Et), and (d) water stress factor ($\beta$) during 2000-2019 summer heatwaves over forested areas (the green region in Figure S2a)".

**Textual comments**

1. L248, caption figure 2 and elsewhere: replace '(total) evaporation' by 'evapotranspiration', whenever you are referring to the sum of transpiration and soil evaporation. Likewise, replace 'recharge' by 'groundwater recharge' if that is what you mean (though it looks like you mean something like 'soil moisture recharge' with this term, which appears odd to me).

On reflection, and differently from our initial response to the reviewer uploaded, we decided to adopt the reviewer's suggestion on the term "evapotranspiration". We changed 'total evaporation' to 'evapotranspiration'.

As noted above, 'recharge' in our paper is not 'groundwater recharge' but 'recharge from the aquifer to the soil column'. We have clarified the recharge is the recharge from aquifer to soil column throughout the paper.

2. L346: 'estimated' > 'estimates'.

Thanks. This has been fixed as,

Line 374-375, "Figure 7 shows the diurnal cycles of $\Delta T$ for the two selected regions (red boxes in Figure 6) compared with the MODIS LST estimates".